# How could phenological records from Chinese poems of the Tang and Song Dynasties (618-1279 AD) be reliable evidence of past climate changes?

Yachen Liu[1], Xiuqi Fang[2], Junhu Dai[3], Huanjiong Wang[3], Zexing Tao[3]

[1]School of Biological and Environmental Engineering, Xi'an University, Xi'an, 710065, China

[2]Faculty of Geographical Science, Key Laboratory of Environment Change and Natural Disaster MOE, Beijing Normal University, Beijing, 100875, China

[3]Key Laboratory of Land Surface Pattern and Simulation, Institute of Geographic Sciences and Natural Resources Research, Chinese Academy of Science (CAS), Beijing, 100101, China

*Correspondence to*: Zexing Tao (taozx.12s@igsnrr.ac.cn)

**Abstract.** Phenological records in historical documents have been proved to be of unique value for reconstructing past climate changes. As a literary genre, poetry reached its peak in the Tang and Song Dynasties (618–1279 AD) in China. Sources from this period could provide abundant phenological records in the absence of phenological observations. However, the reliability of phenological records from poems as well as their processing methods remains to be comprehensively summarized and discussed. In this paper, after introducing the certainties and uncertainties of phenological information in poems, the key processing steps and methods for deriving phenological records from poems and using them in past climate change studies are discussed: (1) two principles, namely the principle of conservatism and the principle of personal experience, should be followed to reduce uncertainties; (2) the phenological records in poems need to be filtered according to the types of poems, background information, rhetorical devices, spatial representations, and human influence; (3) animals and plants are identified at the species level according to their modern distributions and the sequences of different phenophases; (4) phenophases in poems are identified on the basis of modern observation criteria; (5) the dates and sites for the phenophases in poems are confirmed from background information and related studies. As a case study, 86 phenological records from poems of the Tang Dynasty in the Guanzhong Area of China were extracted to reconstruct annual temperature anomalies in specific years in the period between 600–900 AD. Then the reconstruction from poems was compared with relevant reconstructions in published studies to demonstrate the validity and reliability of phenological records from poems in studies of past climate changes. This paper reveals that the phenological records from

poems could be useful evidence of past climate changes after being scientifically processed. This could provide an important reference for future studies in this domain, in both principle and methodology, pursuant of extracting and applying phenological records from poems for larger areas and different periods in Chinese history.

**1 Introduction**

Phenology is the study of recurring biological life cycle stages and the seasonality of non-biological events triggered by environmental changes (Schwartz, 2003; Richardson et al., 2013). Phenological data derived from historical documents have been widely used as proxies to reflect past climatic changes around the world, especially in Europe and Asia. The records of grape harvest dates (Chuine et al., 2004; Meier et al., 2007; Maurer et al., 2009; Daux et al., 2012; Možný et al., 2016;

Labbé et al., 2019), grain harvest dates (Nordli, 2001; Kiss et al., 2011; Wetter and Pfister, 2011; Možný et al., 2012; Pribyl et al., 2012; Brázdil et al., 2018), and ice break-up dates (Tarand and Nordli, 2001; Nordli et al., 2007; Etien et al., 2008) have been adopted to reconstruct past climate changes in Europe. In Japan, cherry blossom records have been used to reconstruct spring temperatures dating back to the medieval period (800–1400 AD) (Aono and Kazui, 2008; Aono and Saito, 2010; Aono,

45    2015).

In China, occasional phenological observations began around 2000 years ago and they have been recorded in various documents. These documents can be further divided into sources produced by institutions and sources generated by individuals. The former includes Chinese classical documents and local gazettes as well as the archives of the Qing Dynasty (1644–1911 AD) and the Republic of China

(1912–1949 AD) (Ge et al., 2008). Based on the documents produced by institutions, abundant phenological records have been extracted to reconstruct past climate changes in specific regions and periods in China (Chu, 1973; Ge et al., 2003; Zheng et al., 2005; Hao et al., 2009; Liu et al., 2016; Fei et al., 2019). However, the phenophases recorded in these documents are mainly non-organic, such as "ice phenology" (the time of freezing and opening of water-bodies), "snow phenology" (the dates of

first and last snowfall) and "frost phenology" (the dates of first and last frosts). The limited amount of organic phenophases which do occur in these documents are principally "agricultural phenology" (e.g., the commencement dates of spring cultivation, winter wheat harvest in summer, and millet harvest in

autumn). Therefore, it is difficult to compare the phenological data from documents produced by institutions with those from modern observations, which largely focus on the seasonal changes of natural plants. In contrast, the phenological information in personal documents (mostly private diaries) is much more varied and includes records about both non-organic and organic events, such as flowers blossoming, leaf expansion and discoloration, and fruit ripening (Ge et al., 2008; Liu et al., 2014; Zheng et al., 2014). Using phenological evidence from diaries, many studies have reconstructed past climate changes in different regions and periods in China (Fang et al., 2005; Xiao et al., 2008; Ge et al., 2014; Wang et al., 2015; Zheng et al., 2018). However, the diaries were most abundant within the past 800 years, especially in the Ming Dynasty (1368–1644 AD), the Qing Dynasty, and the Republic of China. The earliest diary found in China so far (The Diary of Genzi-Xinchou by Lv Zuqian) merely dates back to 1180 AD (Ge et al., 2018). Thus, there is a lack of phenological records on natural plants and animals before the 1180s.

China enjoyed unprecedented economic prosperity, political stability, and a relatively open society in the Tang and Song dynasties (618–1279 AD). The Imperial Examination System, a civil service examination system in imperial China for selecting candidates for the state bureaucracy, had gradually improved and poetry was incorporated into the examination subjects during this period (Zhang 2015). In these contexts, as a literary genre, poetry reached its peak during the Tang and Song Dynasties in ancient China. A very diverse array of people, from emperors to civilians, in the Tang and Song Dynasties preferred to record their thoughts and daily lives in poems. Abundant phenological information that was provided in poems of the Tang and Song Dynasties is a valuable source for phenological records in this period.

Although many studies have indicated that there was a Medieval Warm Period (MWP) in China consistent with many other parts of the world, disputes still exist regarding the start and end time, regional differences, and the extent of warming in different periods of the MWP in China (Zhang et al., 2003; Yang et al., 2007; Ge et al., 2013). The period of the Tang and Song Dynasties coincided with the MWP in China. More reconstruction studies of the Tang and Song Dynasties based on high-resolution proxies will contribute to a better understanding of these controversial issues. Extracting phenological records from poems of the Tang and Song Dynasties is an effective way to improve the resolution of proxy data in this period. However, it is an extraordinary challenge to extract phenological records from poems due to the use of rhetorical devices, the limitations on poetic rules

and forms, as well as the needs of rhymes and sounds in the poems. In addition, the phenological

evidence in poems did not always follow modern criteria, which could lead to considerable

uncertainties if the real phenophases in poems were not properly identified.

Chu (1973) laid the foundation for climate reconstructions based on documents and has been

highly praised worldwide for his efforts. Although a few subsequent studies (Man, 1998; Ge et al.,

2010) adopted phenological evidence from poems to reconstruct climate changes, further systematic

and specialized research on deriving phenological records from poems of the Tang and Song Dynasties

still needs to be carried out.

In this study, we first introduce the characteristics of phenological information in poems,

including its accessibility and inherent uncertainties. We then put forward basic principles and key

processing steps for extracting phenological records from poems of the Tang and Song Dynasties. We

also compare phenological records from poems with other proxies in the reconstruction of past climate

changes in the Guanzhong Area of central China as a case study. Our overall objectives are to

demonstrate the validity and reliability of phenological records from poems as a proxy of past climate

changes and to provide a reference, both theoretical and methodological, for the extraction and

application of phenological records from poems.

**2 The Certainties and Uncertainties of Phenological Information in Poems from the Tang and**
**Song Dynasties**

As a special carrier of historical phenological information, poetry has both certainties and

uncertainties vis-à-vis applications to past climate changes. For example, in the study by Chu (1973),

which laid the foundation for climate reconstructions based on documents, 17 pieces of evidence were

from poems and 11 of them were phenological information of the Tang and Song Dynasties. Most of

the phenological information from poems used by Chu (1973) was valid and the reconstructed results

have been verified by other studies, which demonstrates the certainties of phenological records from

poetry. However, other phenological evidence such as the orange trees in the Guanzhong Area used by

Chu (1973) may be less certain. For instance, some studies have pointed out that the orange trees in the

Guanzhong Area recorded in the poems of the Tang Dynasty (618–907 AD) were transplanted from

other places and were taken care of by specialized personnel in the Imperial Palace (Man 1990; Mu

1996). Therefore, the certainties and uncertainties of phenological information in poems from the Tang and Song Dynasties need to be analyzed before being used in studies of past climate changes.

**2.1 The certainties of phenological information from poems**

Poetry is one of the major genres of Chinese literature. It expresses peoples' social lives and spiritual worlds with concise and emotive words according to the requirements of certain syllables, tones, and rhythms. The poetry of the Tang and Song Dynasties represents the highest level of poetry development and has become a treasure of Chinese traditional literature. People in the Tang and Song Dynasties exhibited a preference for recording and sharing their lives and ideas via poems, which is akin to recording in diaries in the later dynasties. Phenology, which could be used to indicate seasons and guide agricultural activities, is one of the favorite topics of poets. As most of the poems were improvised, they commonly reflect the real-time experiences of the poets. In addition, the great mass of the poems passed down to contemporary times were written by well-educated scholars, who were able to describe the phenological phenomena they saw without misusing or abusing words. Thus, poetry is an excellent carrier of phenological information.

Regarding the different types of poems of the Tang and Song Dynasties, phenological information is most abundant in natural poems and realistic poems. The natural poems describe the force and beauty of nature, such as mountains, rivers, animals, and plants; they contain almost all kinds of phenological records, spanning the organic non-organic (Table 1). The realistic poems strive for the typicality in images, authenticity in details, and objectivity in descriptions. For example, there is a line in a poem by Bai Juyi as follows: "There is a crescent moon on the third night and the cicada sings for the first time" (Appendix D1), which specifically records the phenology of the first call of cicadas. Generally speaking, the phenological information from poems, especially natural poems and realistic poems, is objective and authentic, and can thus be leveraged as a data source for reconstructing past climatic changes.

**2.2 The quantity, spatial distribution, and accessibility of phenological records from the poems**

By their very nature, Chinese poems have many distinctions in terms of recording phenological information compared to documents produced by institutions and personal diaries (Table 2). Poems have evident advantages in the quantity and variety of phenological evidence. According to

Quan-Tang-Shi (the Poetry of the Tang Dynasty) (Peng et al., 1986) and Quan-Song-Shi (the Poetry of the Song Dynasty) (Center for Ancient Classics & Archives of Peking University, 1999), nearly 50 thousand poems from the Tang Dynasty and more than 270 thousand poems from the Song Dynasty are preserved. Numerous phenological records in the poems not only include non-organic events, but also include a variety of organic phenomena, most of which concern the phenology of natural plants and animals.

The spatial distributions of the phenological records are highly consistent with the ruling regions of the dynasties and there is a positive relationship between the quantity of records preserved from particular areas and the level of development of those areas. Take the Song Dynasty (960–1279 AD) as an example: Because north China was dominated by the Jin Dynasty from 1127 to 1279 AD, the phenological records from Quan-Song-Shi of this period are mainly located in southern China, especially around the city Hangzhou (the capital city of the Song Dynasty at that time).

In general, the accessibility of phenological records in poems tends to be lower than that of other documents. Unlike documents produced by institutions in which phenological evidence was recorded by dedicated individuals, the phenological evidence in poems was recorded more inadvertently. The information on phenophases in poems may be incomplete or ambiguous. For a specific phenophase, a poet usually only recorded it a few times in poems during his lifetime. Thus, the frequency and continuity of the phenophase in his poems were relatively low. Take the word "willow" as an example: It was mentioned in 9041 poems in the Quan-Tang-Shi and the Quan-Song-Shi, but clear species names, phenophases, dates, and sites can be obtained from only 80 (0.88%) poems. The accessibility of phenological records in poems may vary depending on particular characteristics of the poets. For example, Li Bai and Du Fu are the most representative romantic poet and realistic poet in the Tang Dynasty, respectively. According to Quan-Tang-Shi, there were 896 poems written by Li Bai and 1158 poems written by Du Fu. Among them, 23 (2.56%) poems by Li Bai and 76 (6.56%) poems by Du Fu are related to phenology. Thus, the accessibility of phenological information from poems by Du Fu is more than twice that of Li Bai. Only by integrating the same phenophase recorded by different poets could improvements be made in terms of frequency and continuity.

**2.3 The inherent uncertainties of phenological evidence in poems**

In addition to the uncertainties arising from data interpretation, calibration, validation, and verification, the extraction of phenological evidence from poems could also be associated with inherent uncertainties during the identification of species, the identification of phenophases, and the ascertainment of dates and sites. Such uncertainties need to be identified as a precursor to using phenological records to reconstruct past climate changes.

**2.3.1 Uncertainties in the identification of species**

Because the Chinese language has not changed fundamentally during the long history of the country, the people in present day China can read ancient poems without too much difficulty. Nevertheless, some changes in meanings and expressions of particular words and phrases still exist. Particular words or phrases may have several additional meanings in ancient Chinese compared with modern usage. For example, the phrase "jin hua" (mainly refers to golden flower in modern Chinese) has at least four meanings in the Quan-Tang-Shi, but only one of them is a substantial description of phenology (Table 3).

The different names of some specific species in ancient China have also been simplified and unified in contemporary language. For example, the Si sheng du juan (*Cuculus micropterus*) had at least three different names during the Tang and Song Dynasties (Table 4). It is also noteworthy that the names of plants and animals in poems were mostly recorded at the genera level due to the lack of modern taxonomic knowledge. Nevertheless, different species within the same genus may exhibit divergent responses to climate change according to modern phenological studies (Dai et al., 2013). Thus, large uncertainties exist during the identification of species in poems.

**2.3.2 Uncertainties in the judgment of phenophases**

Phenophases in poems are not recorded in strict accordance with modern systematic criteria, but are described through multiple rhetorical devices such as metaphor, personification, hyperbole, quote, pun, and rhyme. As such it is difficult to extract clear phenophases from poems. For example, there is a line in a poem by the poet Quan Deyu as follows: "Peonies occupy the spring breeze with their fragrance alone" (Appendix D12), which describes the phase of peonies flowering. However, the phenophase in this line is equivocal due to the use of personification. To compare the phenological

records from poems with corresponding modern observational phenophases, the exact phenological

stages need to be identified from the first flowering date, the full-flowering date, and the end of

flowering date. Therefore, uncertainties may be produced during the identification of specific

phenophases.

### 2.3.3 Uncertainties in ascertainment of dates

Exact dates are crucial for quantitatively evaluating phenological and climatic changes from past

to present. By converting the Chinese lunar calendar into the modern Gregorian calendar, the

phenophases in the poems can be compared with modern observational phenophases. Some poems may

contain precise temporal information. For example, the poet Bai Juyi recorded the following in his

poem: "The azalea is falling and the cuckoo is singing in this year" (Appendix D13). The title of this

poem is "Farewell spring (written on the 30th day of the third month of the 11th year of the Yuan

He)"—Yuan He is one of the reign titles of the Tang Dynasty and the corresponding Gregorian date of

this poem is April 30, 816 AD. However, the time of writing was not explicitly recorded for most other

poems. Any lack of information concerning year, month, or day may lead to failures in phenological

and climatic reconstructions. For instance, in another poem by Bai Juyi he states "People are busy in

the fifth lunar month because the wheat is yellow in the field" (Appendix D14). Here, only information

concerning the month was directly presented in this poem, which could obviously lead to uncertainties

when deducing the year and the day. To make matters worse, some poems were written according to

the memories or imaginations of poets. The information from such poems thus needs to be excluded.

### 2.3.4 Uncertainties in ascertainment of sites

By matching the ancient name of a site with its modern name, the phenophases in poems can be

compared with the corresponding observational phenophases at the same site. However, similar to

dates, the sites of phenophases in poems are sometimes missing. Worse still, some names of sites

mentioned in poems are imagined for the purpose of expressing emotions rather than to record real

locations. For example, Lu You wrote a verse in his poem which reads "There are so many willow

branches in Ba Qiao, but who would have thought sending one to me" (Appendix D15). Ba Qiao is a

location in Xi'an (a city in central China), which is more than 700 km away from the place where Lu

You wrote this poem (Chengdu, China). By describing the willow branches in his hometown in this

poem, the poet expressed his homesickness. When attempting to ascertain sites, these kinds of uncertainties should be carefully considered and dealt with appropriately.

**3 The Methods of Processing Phenological Records in Poems from the Tang and Song Dynasties for Past Climate Studies**

To minimize the uncertainty during the extraction of clear species, phenophases, dates, and sites from poems and to render them comparable with modern observations, several fundamental principles and processing steps should be put forward.

**3.1 The basic principles for data processing**

**3.1.1 The principle of conservatism**

The principle of conservatism refers to deducing ambiguous information conservatively, so as to keep the characteristics of phenological information without causing too much deviation. Take the aforementioned poem of Bai Juyi (Appendix D14) as an example: The poem was written in 807 AD in Xi'an according to background information while the exact date is not recorded. From the poem, we know that the harvest date of wheat in that year appeared in the fifth lunar month (from June 10 to July 8 in the Gregorian calendar), so that June 10 which is the closest to the modern observations (from May 26 to June 8 with the average of June 2) can be determined as the date of wheat harvest in 807 AD in Xi'an. It should be noted that if the recorded period in the poem overlaps with the time of the modern phenophase, the principle of conservatism is inapplicable, and the record in the poem is invalid.

**3.1.2 The principle of personal experience**

The principle of personal experience demands that the phenological information described in the poems was being experienced by the poet, thus excluding records based on imaginations or memories. For example, Yang Wanli recorded a line in his poem which stated that "Begonias in my hometown are flowering on this date and I see them booming in my dream" (Appendix D16). From this, we know that he was not in his hometown when he wrote this poem. Thus, the phenophase of Begonias in this poem cannot be used. It is more complex to diagnose the information in some poems. For example, Lu You wrote a poem in 1208 AD wherein it is recorded that "The Begonias in Biji Fang (place name) are the best in the world. Each branch looks dyed with scarlet blood" (Appendix D17). By looking into the life

experience of Lu You, this poem is found to record his memory in 1172 AD. Therefore, this poem

cannot be used either as phenological evidence according to the principle of personal experience.

### 3.2 The key steps of data processing

On the basis of the foregoing principles, four steps are required for the processing of phenological

records in poems (Figure 1).

### 3.2.1 Step 1: filtering the records

(1) Filtering the records according to the features of poets and poems

Poems commonly reflect the thoughts and daily lives of the poets. Thus, the poems written by

people in certain professions who have little contact with phenological events, such as the alchemists

mentioned in Table 3, may contain little phenological information. In this way, the poems written by

alchemists can be excluded to improve the accessibility of phenological evidence from the poems.

Furthermore, the records can be filtered according to the styles of poems and the interests or life

experiences of the poets. For example, it is more likely that phenological records can be extracted from

pastoral poems than from history-intoned poems.

(2) Filtering the records according to the background information

According to the background information of a poem, we can judge whether the phenophases in the

poem actually happened, thus ensuring the robustness of phenological evidence. For example, there is a

line in a poem by Su Shi as follows: "A few branches of peach blossom outside the bamboo grove, and

the ducks will notice the warming of the river firstly" (Appendix D18). This seems to describe the

natural phenophases in spring. However, by looking into the background information, we know that

this poem refers to a painting. Therefore, it describes the scenery within the painting instead of real

nature. The record thus needs to be excluded.

(3) Filtering the records according to the rhetorical devices

Whether the use of rhetorical devices in poems affects the authenticity of phenophases needs to be

distinguished. For instance, despite the rhetorical device of personification used in the aforementioned

poem by Quan Deyu (Appendix D12), it does reflect the blossom of peonies. Thus, this poem can be

used in the study of past climate changes. The line of Lu Zhaoling saying that "The water in Laizhou

(place name) has become shallower several times and how ripe is the peach fruit" (Appendix D19)

seems to enquire about the time of the peach phenophase, but actually, it is referring to the myth that peaches mature once every three thousand years in wonderland. The rhetorical device of quotation in this line has affected the authenticity of phenophases. Thus, this record should be eliminated.

(4) Filtering the records according to the spatial representations

For a specific species, phenophases vary with latitude, longitude, and elevation. It is necessary to clarify the spatial representation of phenological records in poems and to select records that are not affected by the local microclimate. For example, Bai Juyi recorded in his poem "All the flowers on the plain have withered in the fourth lunar month, but the peaches in the temple on the mountain just begin

to bloom" (Appendix D20). This record cannot be directly compared with modern observational data because the difference in altitude is almost 1000 meters between the mountain in the poem and the modern observation site on the plain. Other factors that contribute to spatial differences such as valleys, depressions, and heat island effects are also used to filter the records.

(5) Filtering the records according to human influence

Human activities, such as cultivation and transplantation could also affect the phenophases of plants. To accurately reflect climate changes, it is necessary to filter the records that were affected by human activities. Take the orange trees in the imperial palace of the Tang Dynasty as an example. Some researchers pointed out that these oranges were transplanted from southern China and could not normally survive the winter on the Guanzhong Plain. Thus, they were intensively managed by humans.

This kind of phenological information in poems cannot be used as an indicator of climate changes.

**3.2.2 Step 2: identifying animals and plants at the species level**

There are two principal ways of identifying the animals and plants in poems from the genera level to the species level. The first way involves identifying the species according to the modern distribution of different species under the genera. For instance, the poet Liu Xian recorded the following

information in his poem: "The flowers of peaches are going to fall while the branches of willow are stretching" (Appendix D21). This poem was written in Xi'an, which is located in the middle reaches of the Yellow River. Historically, the main peach species were *Amygdalus davidiana* and *Amygdalus persica*. According to modern species distributions, the former species can be found along the middle and lower reaches of the Yellow River while the latter occurs in the Huai River basin (Gong et al.,

1983). Thus, the peach in the poem can be identified as *A. davidiana*. The second way is to identify the

species according to the sequences and correlations of different phenophases. For example, Gao Shi wrote a poem in Chengdu wherein it is stated that "The green-up of willow leaves and the ume blossoms can't stop me from being sad" (Appendix D22). The ume plant in ancient Chinese language usually refers to *Chimonanthus praecox* or *Armeniaca mume*. From the content of the text, we can infer that the ume blossoms occurred at a similar time to the leaf expansion of willow. According to modern observation data in Chengdu, the average full leaf expansion date of willow (*Salix babylonica*) is February 23, while the average full flowering dates of *Chimonanthus praecox* and *Armeniaca mume* are January 10 and February 10, respectively. The average date of full flowering for *A. mume* is closer in time with the average date of full leaf expansion for willow. Thus, the ume blossoms in the poem can be identified as *A. mume*.

### 3.2.3 Step 3: identifying the phenophases according to modern observation criteria

By applying the semantic differential technique, which is commonly used in the studies of past climate changes (Academy of Meteorological Science of China Central Meteorological Administration, 1981; Wang, 1991; Wei et al., 2015; Yin et al., 2016; Su et al., 2018; Fang et al., 2019), the descriptions in poems are classified and graded according to the criteria of the phenological observation methods in China (Wan and Liu, 1979; Gong et al., 1983; Fang et al., 2005). Taking the aforementioned poem of Quan Deyu (Appendix D12) as an example, a line describes a scene where many peonies were blooming and filling the spring breeze with strong perfume. By classifying and grading the key words "occupy" and "fragrance" in this poem with other common descriptions of flowering phases in poems such as "tender", "sparse", "flourish", "dense", "wither", and "fallen", the description of peony blooming in this poem was most likely to match with the full flowering date under the modern criterion "more than half of the flowers have blossomed in the observed species". Thus, the phenophase in the poem can be identified as the full flowering date. The classification and grading results for some representative examples of phenological descriptions in poems are shown in Table 5.

### 3.2.4 Step 4: ascertaining the dates and locations

This step firstly sought temporal information, including clear year, month, and date of the phenophase, from the titles, prefaces, and lines of the poems. The missing time information could then be deduced by consulting the background information, related studies or estimated reasonably

according to the principle of conservatism. Finally, the time information in the Chinese lunar calendar had to be converted into the modern Gregorian calendar. For example, the poet Cui Riyong recorded in his poem "The ume blossoms in the palace smell fragrant and look delicate with the background of snow" (Appendix D29). The title of this poem indicates that this poem records a banquet in the imperial palace on People's Day (Chinese traditional festival on the 7th day of the first lunar month). From the poem, we do not know which year it was. However, this banquet was also recorded by Xin Tang Shu (New Books of Tang, a history book of the Tang Dynasty) in the year 730 AD. Hence, we know that this poem was written in 730 AD.

Similarly, the exact location of the sites could be confirmed. It should be checked whether the place names appearing in the poems are real sites for phenophases. For example, Ba qiao is not the site of the phenophase for willow in the aforementioned poem by Lu You (Appendix D15). Thus, the record in this poem cannot be used as phenological evidence for past climate studies.

**4 Validation of the phenological records from poems for reconstructing past climate changes: A case study of temperature reconstruction in the Guanzhong Area for specific years during 600–900 AD**

To test the reliability of phenological records in poems for past climate change studies and the validity of the processing methods outlined in this study, we extracted 86 phenological records (Appendix A) from poems of the Tang Dynasty to reconstruct the mean annual temperatures in the Guanzhong Area of China during the period of 600–900 AD.

**4.1 Study area**

The Guanzhong Area (33°35′–35°50′N, 106°18′–110°37′E), located in central China (Figure 2), was where the capital city of the Tang dynasty was located. Many poets were active here and left many poems describing phenology during the Tang dynasty. The study area has a continental monsoon climate with mean annual temperatures ranging from 7.8 ℃ to 13.5 ℃ and mean annual precipitation from 500 mm in the northeast to 700 mm in the southwest (Qian, 1991).

**4.2 Data and methods**

Since the 86 records from poems pertain to diverse phenophases, they indicate temperature changes at different times of the year. To obtain a relatively uniform and comparable series of

reconstructed temperatures, the mean annual temperature anomaly was selected as the reconstruction index. Transfer functions between annual temperature anomalies and corresponding phenophases were established by using modern observation data. The transfer functions were then applied to reconstruct the annual temperature anomalies (with the reference period of 1961–1990 AD) in the Guanzhong Area during 600–900 AD. The modern phenological and meteorological data used and the detailed methods of the transfer functions are shown in Appendix B.

## 4.3 Results and comparisons with other reconstructions

Figure 3(a) shows the reconstructed annual temperature anomalies using the phenological records from poems. For validation purposes, the results were compared with relevant studies. The first series used for comparison is attributable to Liu et al (2016), wherein winter half-year (from October to next April) temperature anomalies were reconstructed by 87 phenological records from historical documents (mostly produced by institutions) for the period 600–902 AD in the Guanzhong Area. The reconstruction by Liu et al (2016) is a reliable reference not only because of the study area and period considered coincide, but also because the proxies used by that study and ours are phenological records from independent sources. To avoid the additional influences of reconstruction indicators and transfer functions, the records from Liu et al (2016) were reconstructed to annual temperature anomalies (Figure 3(b)).

Table 6 shows the historical data sources, types, and quantity of phenological evidence in Liu et al. (2016) and this study. Except for a single record in one poem (Appendix A13), there is no duplication in records between the two studies. In general, the two studies are based on similar quantities of evidence, while the data types used in the two studies are quite different. In terms of Liu et al. (2016), 71 of 87 (nearly 82%) pieces of phenological data are from documents produced by institutions. Among the 87 pieces of evidence, 67 (more than 77%) are non-organic phenophases or agricultural phenophases (Figure 3(b)). By contrast, the vast majority (more than 96%) of evidence from poems in this study are phenophases of wild plants (Figure 33(a)). These differences suggest that the phenological records in poems are effective supplements to historical phenological evidence for the period of the Tang Dynasty. It is also worth noting that fewer years are reconstructed in this study (36) compared to in Liu et al. (2016) (76), which further supports the claim that the frequency and

continuity of phenological records preserved in poems is more sporadic than that of documents produced by institutions (Table 2).

To assess the validity of the temperature reconstruction from poems, two more temperature reconstructions by different proxies were leveraged for comparison. The first was winter half-year temperature anomalies at a 30-year resolution reconstructed from documentary evidence in the middle
and lower reaches of the Yellow and Yangtze Rivers of China (Ge et al., 2003) (Figure 3(c)). The second was annual temperature anomalies reconstructed from tree rings in Asia (Ahmed et al., 2013) (Figure 3(d)). All four reconstructions indicated that there were more relatively cold years in the later periods after around the 800s. Indeed, the coldest years according to all four reconstructions occurred in this period. Before the 800s, the reconstructions by Liu et al. (2016), Ge et al. (2003), and our study showed
more relatively warm temperatures with the warmest years occurring around the 660s. Furthermore, the amplitude of reconstructed temperature by Liu et al. (2016) was 3.30 ℃, which was very similar to the amplitude of reconstructed temperature (3.28 ℃) in our study. As a benchmark, according to modern data spanning 1951–2013 the amplitude was 3.97 ℃. In sum, the similarities between different reconstructions confirm the effectiveness of phenological records from poems for gauging past climate
changes.

**5 Discussions**

There are still controversies on how the climate changed during the Tang and Song Dynasties (Chu, 1973; Fei et al., 2001; Yang et al., 2002; Ge et al., 2003; Tan et al., 2003; Thompson et al., 2006; Zhang and Lu, 2007). One of the reasons lies in the lack of sufficient evidence supporting the climatic
reconstructions. Although some studies have reconstructed the temperatures during this period using natural evidence such as tree rings, pollens, and sediments (Xu et al., 2004; Zhang et al., 2014; Zhu et al., 2019), their results either do not cover the entire period or they have relatively low temporal resolution. In addition, these natural proxies are mostly collected from uninhabited areas, thus they are not particularly amenable to evaluating the interactions between climate change and human activities.
In comparison, documentary evidence, which occurs more frequently and is closer to human life, has become an important data source for reconstructing climate changes in this period. As one of the most popular literary forms in the Tang and Song Dynasties, poetry has huge potential to provide abundant

and diverse phenological information, which will undoubtedly contribute to the study of historical climate change.

Despite this, very few studies have thus far been reported concerning the use of phenological records from poems to quantitatively reconstruct historical climate change due to the lack of effective methodologies for data extraction. Unlike climate reconstructions using other proxies that have standard processing methods and clear reference objects, the processing of phenological records from poems is much more complex. For example, dating tree-ring samples only requires counting the

number of annual rings from the outside to the inside or comparing them with a standard chronology. However, the temporal information in poems cannot be obtained directly from a reference chronology. As already mentioned, the temporal information in poems may be hidden in the poet's biography, the official history book, or some related studies. It is necessary to search through these materials one by one and make careful comparisons before ascertaining the exact temporal information, and indeed

some information is found to be unrecorded after searching through large amounts of material. This problem also exists when seeking to extract information concerning species, phenophases, and sites from poems.

We attempt to introduce a standard procedure for extracting phenological records from poems, which could, on the one hand, minimize the uncertainties of the records, and on the other hand,

efficiently filter irrelevant records. By following the principles and steps herein, researchers can understand where to find the information needed and how to manage the phenological data from poems. The extracted phenological records are comparable with modern observation data and can be used as a proxy for quantitatively reconstructing climate changes.

Although the validity of phenological records from poems has only been tested in a single area of

China in the Tang Dynasty, the methodologies of extracting and processing phenological records from poems for climate reconstructions proposed in this study could be applied to wider regions and longer periods. On the one hand, many studies have demonstrated that climate is the primary driver of phenophases in the whole of China (Piao et al., 2006; Dai et al., 2014; Ge et al., 2015; Tao et al., 2017), which indicates that the phenological records obtained at any place could be used as evidence of

climate changes. On the other hand, historians agree that the feudal society in Chinese history did not fundamentally change during different dynasties (Liu, 1981; Tian, 1982; Feng, 1994). Although historical China varied its administrated area coverage from dynasty to dynasty, its core

socio-economics closely aligned with the major agricultural areas throughout history. This geographic and temporal overlap allows for continuous comparison across the core areas of China (Fang et al., 2019). Correspondingly, the essence of literature, especially poetry, has not changed, though different types of poetry varied in their popularity between dynasties e.g., differences in terms of poetic forms, the number of words, and the needs of rhymes and sounds. Therefore, the phenological records obtained from poems from different periods in core areas of historical China can also be extracted and processed for climatic reconstruction according to the method in this study.

We only used 86 phenological records extracted from poems to reconstruct the temperature anomalies for a small area in the Tang Dynasty. Although the uncertainties from transfer functions are shown in Appendix C, there are other uncertainties that are difficult to quantitatively assess. For example, differences in cultivated plant types and crop management may have an effect on the temperature reconstruction, though many studies show that phenological changes in cultivated plants are principally driven by climate changes, especially temperature variations (Estrella et al., 2007; Lobell et al., 2012; Liu et al., 2018). Overall, the reconstruction in this study testifies to the reliability of phenological records from poems in indicating past climate changes. Nevertheless, there are still many phenological records which remain to be extracted. By rough estimation, the temporal resolution of the phenological records from poems of the Tang and Song Dynasties can reach 20 years or less. In addition, phenological records from poems of the Tang and Song Dynasties are widely distributed, covering almost all the regions of modern China. Take the Song Dynasty (960–1279 AD) as an example. Although north China was dominated by the Jin Dynasty from 1127 to 1279 AD, which means that most poems written by poets living in north China are not contained in the Quan-Song-Shi, we can try to search from the Quan-Jin-Shi (the Poetry of the Jin Dynasty) (Xue and Guo, 1995) to add phenological records in north China. The rich records around the capitals and developed cities are of great value in terms of comparisons with modern phenological observations. Future work will be focused on extracting more records from poems, and developing integration methods for different phenophases at different sites to explore the overall phenological changes and climate changes over a larger region.

 **6 Conclusions**

In this study, we put forward a processing method to extract phenological information from poems of the Tang and Song Dynasties, which includes two principles (the principle of conservatism and the principle of personal experience) and four steps: (1) filtering the records based on the features of poets and poems, the background information, the rhetorical devices, the spatial representations, and human influence; (2) identifying the animals and plants to the species level; (3) judging the phenophases according to modern observation criteria; (4) ascertaining times and sites. We then used this method to extract 86 phenological records from poems of the Guanzhong Area in central China and reconstructed the annual mean temperature anomalies for specific years during 600–900 AD. The reconstructed temperature anomaly series was comparable with that reconstructed by records from documents in the same area and period, demonstrating that our method is effective and reliable. This paper therefore provides a reference in both theory and method for the extraction and application of phenological records from poems in studies of past climate changes.

**Data availability.**

All the data used to perform the analysis in this study are described and properly referenced in the paper. The phenological records from poems used to reconstruct the annual temperatures are listed in Appendix A and all the original and sources of the verses used in this paper are listed in Appendix D in Chinese. The modern phenological data are available from the National Earth System Science Data Center(2020). The modern meteorological data are available from the China Meteorological Data Service Center (2020).

**Author contributions.**

Yachen Liu and Zexing Tao contributed to the idea and design of the structure of paper; Yachen Liu collected and analyzed the data; Yachen Liu, Xiuqi Fang, Junhu Dai, Huanjiong Wang and Zexing Tao wrote the paper.

**Competing interests.**

The authors declare that they have no conflict of interest.

**Special issue statement.**

This article is part of the special issue "International methods and comparisons in climate reconstruction and impacts from archives of societies".

**Acknowledgements.**

We would like to thank the anonymous reviewers and editors for their valuable comments.

**Financial support.**

This study was supported by the National Natural Science Foundation of China (41807438, 41771056), the Strategic Project of Science and Technology of the Chinese Academy of Sciences (XDA19040101), the National Key R & D Program of China (2018YFA0606102), and the Special Scientific Research Program of Education Department of Shaanxi Provincial Government (20JK0877).

**Review statement.**

This paper was edited by Qing Pei and reviewed by three anonymous referees.

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

**Figures and tables**

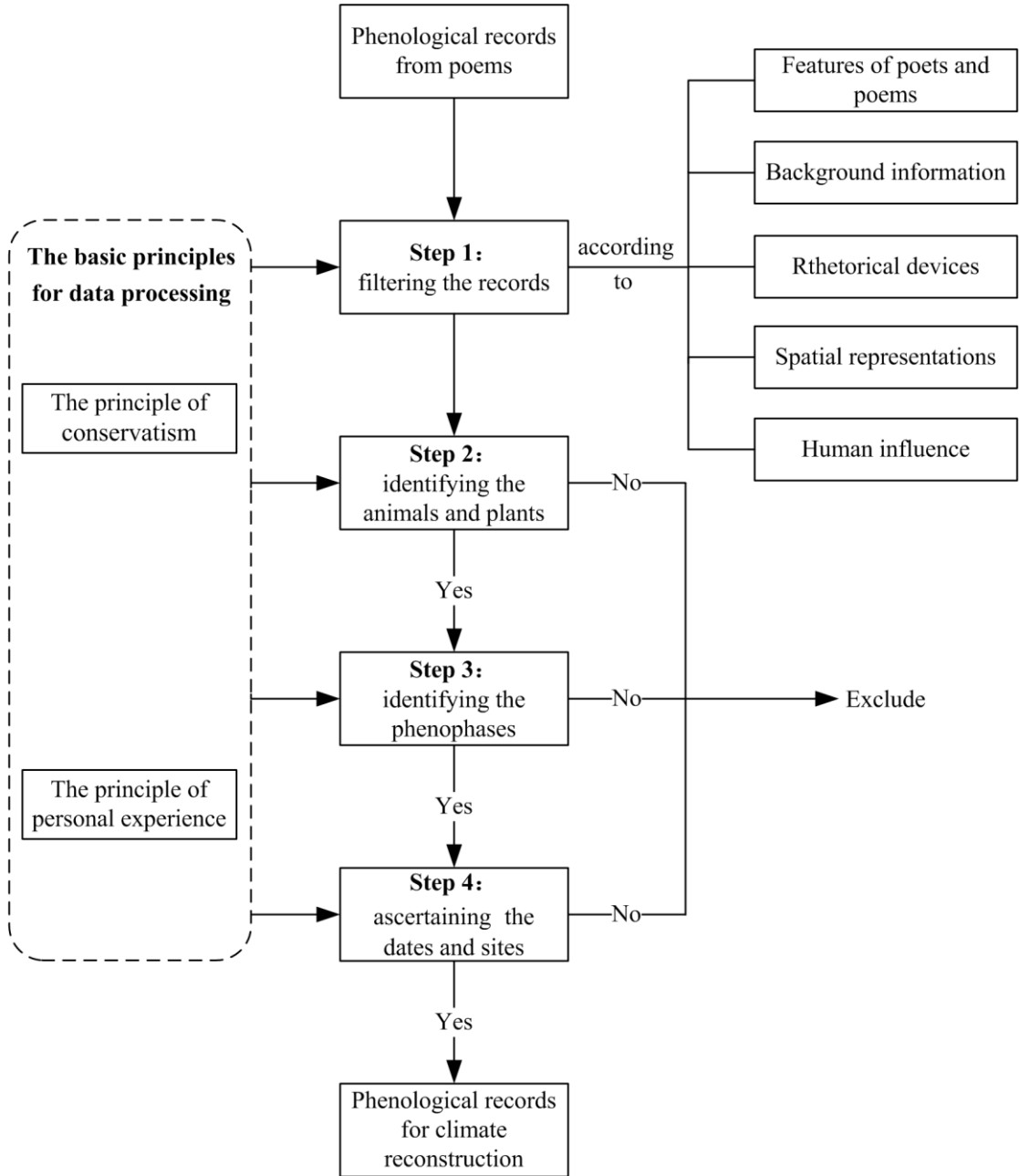

**Figure 1 Processing steps of phenological records in poems for climate reconstructions.**

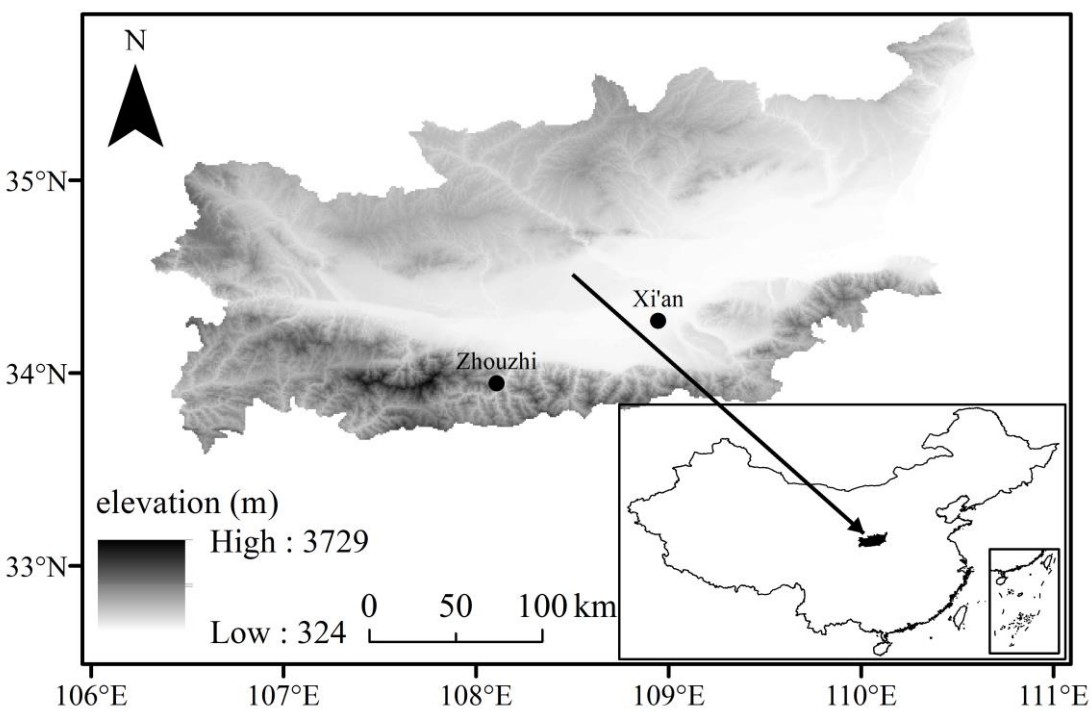

**Figure 2 The location of the Guanzhong Area for the climatic reconstructions in this study with the modern names of sites mentioned in the poems.**

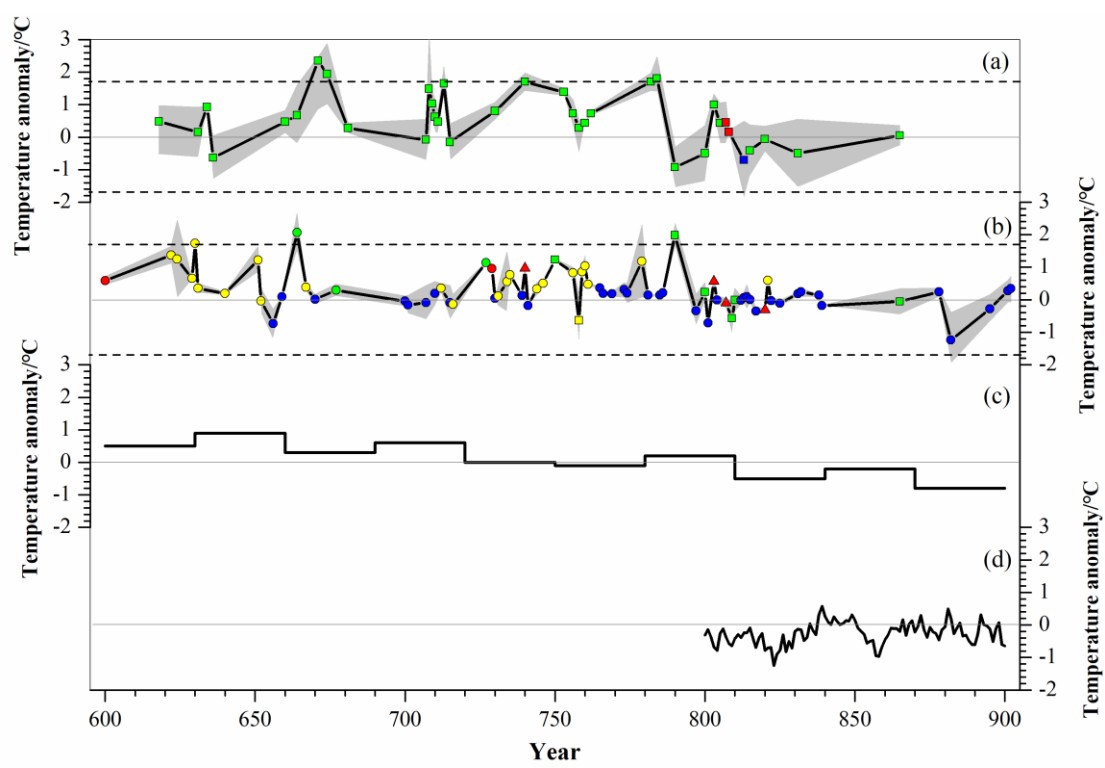

**Figure 3 Comparison of reconstructed temperature anomalies for 600–900 AD (with respect to the mean climatology between 1961 and 1990).** (a): Annual mean temperature anomalies reconstructed

by phenological records from poems in this study; (b): Annual mean temperature anomalies reconstructed using the phenological records from historical documents by Liu et al. (2016); (c): Winter half-year temperature anomalies reconstructed from historical documents for the middle and lower reaches of the Yellow and Yangtze Rivers with a 30-year temporal resolution by Ge et al. (2003). (d): Annual mean temperature reconstructed from tree rings for the whole of Asia by Ahmed et al. (2013). Squares denote temperature anomalies reconstructed from poems; circles denote temperature anomalies reconstructed from documents of institutions; triangles denotes temperature anomalies reconstructed from both poems and documents of institutions; green denotes temperature anomalies reconstructed by phenophases of wild plants; yellow denotes temperature anomalies reconstructed by agricultural phenophases; blue denotes temperature anomalies reconstructed by non-organic phenophases; red denotes temperature anomalies reconstructed by at least two types of phenophases; the gray area approximates the 95% confidence interval according to ordinary least squares (OLS) regression; the dotted lines indicate the 2 standard deviation range of 1.72 °C of the modern period (1951–2013).

**Table 1 Different types of phenology in poems from the Tang and Song Dynasties.**

| Types of phenology | | Example quotes from poems |
| --- | --- | --- |
| Non-organic | phenology of ice | All the springs are frozen and stagnant (Appendix D2). |
| | phenology of snow | It snows in the 8th lunar month in frontier regions (Appendix D3). |
| | phenology of frost | Frost falls in the 8th lunar month of every year (Appendix D4). |
| Organic | phenology of agriculture | The people have just finished planting mulberry trees to raise silkworms and they are going to transplant rice seedling again (Appendix D5). |
| | phenology of natural plants | Ume blossoms begin to bloom in early winter (Appendix D6). |
| | phenology of animals | The river reflects the autumn scenery and the geese begin to fly south (Appendix D7). |

**Table 2 Comparisons among the phenological evidence from poems, diaries, and documents**

**produced by institutions in China.**

| | Poems | Diaries | Documents produced by institutions |
|---|---|---|---|
| Types of phenological evidence | organic (phenology of plants and animals) and non-organic (phenology of ice, snow, and frost) | organic (phenology of plants and animals) and non-organic (phenology of ice, snow, and frost) | mostly non-organic (phenology of ice, snow and, frost) and a few organic (agricultural phenology) |
| Amount of phenological evidence | more | More | less |
| Reasons for phenological record-keeping | memory of daily life/expressing feelings | memory of daily life/observing phenology | recording extreme climatic events and agriculture-related activities |
| Frequency of phenological record-keeping | sporadic | sporadic to phenophase-specific recurrent | phenophase-specific recurrent |
| Continuity of phenological record-keeping | intermittent | intermittent/less than the lifetime of the observer | up to the occurrence of extreme climatic events |
| Species clarity | ambiguous to species-specific | ambiguous to species-specific | most clear |
| Phenophases clarity | ambiguous to phenophase-specific | ambiguous to phenophase-specific | most clear |
| Spatial clarity | ambiguous to inferable | clear/inferable | most clear |

| Temporal clarity | ambiguous to inferable | clear/inferable | most clear |
|---|---|---|---|

**Table 3 Different meanings of the Chinese phrase "jin hua" in poems of the Tang Dynasty.**

| Pinyin of the verse | Meanings of "jin hua" in the poems |
|---|---|
| fan ci huang **jin hua** (Appendix D8) | chrysanthemum (inferred from context) |
| sheng li **jin hua** qiao nai han (Appendix D9) | decorations on ladies' headwear |
| xuan miao mei **jin hua** (Appendix D10) | an alchemistic term for Taoist priests |
| cui wei **jin hua** bu ci ru (Appendix D11) | golden patterns on the tails of peacocks |


**Table 4 Comparisons among the ancient, modern, and Latin names of several common species.**

| Species | Pinyin of ancient name | Pinyin of modern names | Latin name |
|---|---|---|---|
| Animals | Si jiu, Zi gui, Du yu | Si sheng du juan | *Cuculus micropterus* |
| | Cang geng, Shang geng, Chu que, Huang niao | Hei zhen huang li | *Oriolus chinensis* |
| | Xuan niao, Yi niao, Luan niao, Tian nv, Wu yi | Jia yan | *Hirundo rustica* |
| | Tiao, Fu yu, Ni, Qi nv | Cao chan | *Mogannia conica* |
| Plants | Fu qu, Fu rong, Han dan | Lian | *Nelumbo nucifera* |
| | Lu, Wei, Jian jia | Lu wei | *Phragmites australis* |
| | Shan shi liu, Ying shan hong, Shan zhi zhu | Du juan | *Rhododendron simsii* |
| | Mu li, ming zha, Man zha | Mu gua | *Chaenomeles sinensis* |


**Table 5 Classification and grading results for representative examples of phenological descriptions in poems.**

| Phenophase | Translation of the original verses | Description in the modern observation |
|---|---|---|

|  |  | criteria |
| --- | --- | --- |
| First song | New cicada tweeted two or three times (Appendix D23) | The date of first call |
| First appearance | New swallow came ten days before the festival of She (Appendix D24) | The date of first appearance |
| First leaf | Willow leaves are tender just like a beauty frown slightly (Appendix D25) | The date when the first one or two leaves are spread out |
| Full leaf expansion | The green lotus leaves stretch to the horizon (Appendix D26) | The date when the leaflets on half of the branches of the observed tree are completely flat |
| First flowering | The hibiscus is at the beginning of the red and they cover the palace (Appendix D27) | The date when the petals of one or several flowers begin to open fully |
| Full flowering | Peonies occupy the spring breeze with their fragrance alone (Appendix D12) | The date when more than half of the flowers have blossomed in the observed species |
| End of flowering | The flowers of peach are going to fall while the branches of willow are stretching (Appendix D21) | The date when there are very few flowers on the observed trees |
| Fruit drop | The willows and poplars in the street are shrouded in smog (Appendix D28) | The date when *Salix* spp. and *Populus* spp. begin to have fluffy catkins |

**Table 6 Comparisons of data sources, types, and numbers of records used in Liu et al. (2016) and in this study.**

|  | Liu et al. (2016) | | | This study |
| --- | --- | --- | --- | --- |
|  | Documents of institutions | Poems | Total | Poems |
| Non-organic phenophases | 42 | 0 | 42 | 1 |
| Agricultural phenophases | 24 | 1 | 25 | 1 |

| | | | | |
|---|---|---|---|---|
| Phenophases of natural plants | 5 | 15 | 20 | 83 |
| Phenophases of animals | 0 | 0 | 0 | 1 |
| Total | 71 | 16 | 87 | 86 |

**Appendix A: Phenological records from poems used in the reconstruction of this study.**

| No. | Gregorian date | Site | Phenophase | Translation of the original verses |
|---|---|---|---|---|
| 1 | 28 June 618 | Xi'an | End flowering date of *Punica granatum* | It missed the spring because of late blooming (Appendix D30). |
| 2 | 27 February 631 | Xi'an | Full leaf expansion date of *Salix babylonica* | The leaves of willow welcome the third lunar month and the ume blossoms take the two years apart (Appendix D31). |
| 3 | 27 February 631 | Xi'an | Full-flowering date of *Armeniaca mume* | The leaves of willow welcome the third lunar month and the ume blossoms take the two years apart (Appendix D31). |
| 4 | 18 January 634 | Xi'an | Full-flowering date of *Chimonanthus praecox* | There are no leaves on the willow tree, but flowers on the ume tree (Appendix D32). |
| 5 | 27 April 636 | Xi'an | Full-flowering date of *Juglans regia* | Peach flowers blossom for those who are going away (Appendix D33). |
| 6 | 10 September 660 | Xi'an | Full-flowering date of *Osmanthus fragrans* | Only osmanthus blooms near the south hill (Appendix D34). |
| 7 | 31 August 664 | Xi'an | End flowering date of *Osmanthus fragrans* | Osmanthus is at the end of flowering in the moonlight and the ume tree is at the beginning of flowering under the beam |

| | | | | |
|---|---|---|---|---|
| | | | | (Appendix D35). |
| 8 | 31 August 664 | Xi'an | First flowering date of *Chimonanthus praecox* | Osmanthus is at the end of flowering in the moonlight and the ume tree is at the beginning of flowering under the beam (Appendix D35). |
| 9 | 8 February 671 | Xi'an | First flowering date of *Armeniaca mume* | Ume blossoms early in the palace and the willow is new near the creek (Appendix D36). |
| 10 | 8 February 671 | Xi'an | First leaf date of *Salix babylonica* | Ume blossoms early in the palace and the willow is new near the creek (Appendix D36). |
| 11 | 18 February 674 | Xi'an | Full leaf expansion date of *Salix babylonica* | The wicker swings to show its beauty (Appendix D37). |
| 12 | 11 August 681 | Xi'an | Fruit maturity date of *Amygdalus davidiana* | The peaches in the palace are very luxuriant (Appendix D38). |
| 13 | 6 April 707 | Xi'an | End flowering date of *Amygdalus davidiana* | The flowers of peach are going to fall while the branches of willow are stretching (Appendix D21). |
| 14 | 6 April 707 | Xi'an | Full leaf expansion date of *Salix babylonica* | The flowers of peaches are going to fall while the branches of willow are stretching (Appendix D21). |
| 15 | 4 February 708 | Xi'an | First leaf date of *Salix babylonica* | The delicate wicker on the embankment has not turned yellow (Appendix D39). |
| 16 | 4 February 708 | Xi'an | First flowering date of *Armeniaca mume* | The fragrance of ume blossoms and the color of willows can withstand praise (Appendix D40). |
| 17 | 4 February 708 | Xi'an | First leaf date of *Salix* | The fragrance of ume blossoms and |

| | | | | babylonica | the color of willows can withstand praise (Appendix D40). |
|---|---|---|---|---|---|
| 18 | 4 February r 708 | Xi'an | First flowering date of *Armeniaca mume* | The fragrance of ume blossoms seems to be obscured by beautiful singing (Appendix D41). |
| 19 | 4 February 708 | Xi'an | First flowering date of *Armeniaca mume* | Ume blossoms vie to bloom in the palace (Appendix D42). |
| 20 | 4 February 708 | Xi'an | First flowering date of *Armeniaca mume* | The ume blossoms and willows in the palace can recognize the weather (Appendix D43). |
| 21 | 4 February 708 | Xi'an | First leaf date of *Salix babylonica* | The ume blossoms and willows in the palace can recognize the weather (Appendix D43). |
| 22 | 4 February 708 | Xi'an | First flowering date of *Amygdalus davidiana* | Why do peaches and plums compete to bloom (Appendix D44). |
| 23 | 4 February 708 | Xi'an | First flowering date of *Prunus salicina* | Why do peaches and plums compete to bloom (Appendix D44). |
| 24 | 4 February 708 | Xi'an | First flowering date of *Armeniaca vulgaris* | New apricot blossoms adorn the palace and ume blossoms bloom at the feast (Appendix D45). |
| 25 | 4 February 708 | Xi'an | First flowering date of *Armeniaca mume* | New apricot blossoms adorn the palace and ume blossoms bloom at the feast (Appendix D45). |
| 26 | 10 February 709 | Xi'an | Full-flowering date of *Chimonanthus praecox* | The flicking of snow on the branches adds to the beauty of ume blossoms (Appendix D46). |
| 27 | 21 February 709 | Xi'an | First flowering date of *Armeniaca mume* | Ume blossoms and willow catkins are new (Appendix D47). |
| 28 | 15 March 709 | Xi'an | Full leaf expansion date of | The willows leaves are all open |

| | | | | over the city (Appendix D48). |
|---|---|---|---|---|
| | | | *Salix babylonica* | |
| 29 | 15 March 709 | Xi'an | Full leaf expansion date of *Salix babylonica* | Willows secretly urge the late spring (Appendix D49). |
| 30 | 17 April 709 | Xi'an | Beginning date of fruit drop of *Salix babylonica* | The willow by the river flicks the emperor's goblet (Appendix D50). |
| 31 | 16 October 709 | Xi'an | End flowering date of *Osmanthus fragrans* | The osmanthus fell into the goblet full of wine (Appendix D51). |
| 32 | 4 March 710 | Xi'an | Full-flowering date of *Armeniaca mume* | The ume blossoms remain white when the cold is over while the willows have not turned yellow when the wind is late (Appendix D52). |
| 33 | 4 March 710 | Xi'an | Full leaf expansion date of *Salix babylonica* | The ume blossoms remain white when the cold is over while the willows have not turned yellow when the wind is late (Appendix D52). |
| 34 | 4 March 710 | Xi'an | Full leaf expansion date of *Salix babylonica* | There are thousands of willows unfolding their leaves (Appendix D53). |
| 35 | 25 March 710 | Guanzhong | Full-flowering date of *Amygdalus davidiana* | There are red flowers all over the ground and the whole banquet is filled with fragrance (Appendix D54). |
| 36 | 25 March 710 | Guanzhong | Full-flowering date of *Amygdalus davidiana* | The red calyxes bloom against the dawn in the garden (Appendix D55). |
| 37 | 25 March 710 | Guanzhong | Full-flowering date of *Amygdalus davidiana* | The peach blossoms are bright and seem to have brilliance (Appendix |

| | | | | D56). |
|---|---|---|---|---|
| 38 | 25 March 710 | Guanzhong | Full-flowering date of *Amygdalus davidiana* | Countless flowers bloom among the flowers by the water (Appendix D57). |
| 39 | 25 March 710 | Guanzhong | Full-flowering date of *Amygdalus davidiana* | The gorgeous flowers in the garden accompany the beauty (Appendix D58). |
| 40 | 3 April 710 | Guanzhong | End flowering date of *Amygdalus davidiana* | The peach blossoms by the Wei River fall into the water (Appendix D59). |
| 41 | 4 April 710 | Xi'an | Full-flowering date of *Amygdalus davidiana* | When the peaches and plums bloom in spring, the scenery of the capital city is good (Appendix D60). |
| 42 | 4 April 710 | Xi'an | Full-flowering date of *Prunus salicina* | When the peaches and plums bloom in spring, the scenery of the capital city is good (Appendix D60). |
| 43 | 4 April 710 | Xi'an | Beginning date of fruit drop of *Salix babylonica* | The red calyx exudes fragrance and the branches of willows are surrounded by green ribbons (Appendix D61). |
| 44 | 4 April 710 | Xi'an | Full-flowering date of *Amygdalus davidiana* | The red calyx exudes fragrance and the branches of willows are surrounded by green ribbons (Appendix D61). |
| 45 | 5 April 710 | Xi'an | End flowering date of *Armeniaca mume* | The ume blossoms in the palace glowed against the snow and the willow trees in the city were full of smog (Appendix D62). |
| 46 | 5 April 710 | Xi'an | Beginning date of fruit | The ume blossoms in the palace |

| | | | | |
|---|---|---|---|---|
| | | | drop of *Salix babylonica* | glowed against the snow and the willow trees in the city were full of smog (Appendix D62). |
| 47 | 5 April 710 | Xi'an | Beginning date of fruit drop of *Salix babylonica* | The willows and ume blossoms in the palace are covered with green ribbons (Appendix D63). |
| 48 | 5 April 710 | Xi'an | End flowering date of *Armeniaca mume* | The willows and ume blossoms in the palace are covered with green ribbons (Appendix D63). |
| 49 | 5 April 710 | Xi'an | Beginning date of fruit drop of *Salix babylonica* | The willows are covered with green smog (Appendix D64). |
| 50 | 6 April 710 | Xi'an | Beginning date of fruit drop of *Salix babylonica* | The green ribbons from the willows float at the banquet (Appendix D65). |
| 51 | 6 April 710 | Xi'an | End flowering date of *Amygdalus davidiana* | Red peach blossoms and emerald green willows adorn the fete (Appendix D66). |
| 52 | 6 April 710 | Xi'an | Beginning date of fruit drop of *Salix babylonica* | Red peach blossoms and emerald green willows adorn the fete (Appendix D66). |
| 53 | 9 May 710 | Xi'an | First flowering date of *Hibiscus syriacus* | Trees cover the palace and the hibiscuses start to turn red (Appendix D67). |
| 54 | 24 March 711 | Guanzhong | Full-flowering date of *Prunus salicina* | The peach and plum blossoms are lost in their own fragrance (Appendix D68). |
| 55 | 24 March 711 | Guanzhong | Full-flowering date of *Amygdalus davidiana* | The peach and plum blossoms are lost in their own fragrance (Appendix D68). |

| 56 | 14 February 713 | Xi'an | End flowering date of *Chimonanthus praecox* | The garden is only accompanied by withered ume blossoms in spring (Appendix D69). |
|---|---|---|---|---|
| 57 | 28 February 713 | Xi'an | First leaf date of *Salix babylonica* | The branches of willows are fresh (Appendix D70). |
| 58 | 7 April 715 | Xi'an | End flowering date of *Amygdalus davidiana* | The pool water is covered with peach blossoms (Appendix D71). |
| 59 | 29 January 730 | Xi'an | Full-flowering date of *Chimonanthus praecox* | The ume blossoms in the palace smell fragrant and look delicate with the background of snow (Appendix D29). |
| 60 | 3 April 740 | Xi'an | Beginning date of fruit drop of *Salix babylonica* | People at the banquet all resent the falling catkins (Appendix D72). |
| 61 | 10 April 753 | Xi'an | Beginning date of fruit drop of *Salix babylonica* | The catkins fall like snowflakes (Appendix D73). |
| 62 | 5 February 756 | Xi'an | Full-flowering date of *Chimonanthus praecox* | The umes bloom towards the sky (Appendix D74). |
| 63 | 18 March 758 | Xi'an | First leaf date of *Salix babylonica* | There are thousands of tender branches of willows in the palace (Appendix D75). |
| 64 | 18 March 758 | Xi'an | Full-flowering date of *Amygdalus davidiana* | Peach blossoms are as red as drunk (Appendix D76). |
| 65 | 15 April 758 | Xi'an | End flowering date of *Amygdalus davidiana* | The peach blossoms wither after the catkins (Appendix D77). |
| 66 | 15 April 758 | Xi'an | Beginning date of fruit drop of *Salix babylonica* | The peach blossoms wither after the catkins (Appendix D77). |
| 67 | 3 April 760 | Xi'an | Full-flowering date of *Pyrus betulaefolia* | Pear flowers bloom during the Cold Food Festival (Appendix D78). |
| 68 | 18 March 762 | Xi'an | Full leaf expansion date of | Flowers and willows in every |

| | | | *Salix babylonica* | village bloom of their own accord (Appendix D79). |
|---|---|---|---|---|
| 69 | 3 April 782 | Xi'an | Beginning date of fruit drop of *Salix babylonica* | In spring the city is full of flying catkins (Appendix D80). |
| 70 | 25 February 784 | Xi'an | First leaf date of *Salix babylonica* | The flowers and willows in the capital are fresh (Appendix D81). |
| 71 | 19 April 790 | Xi'an | Full-flowering date of *Paeonia suffruticosa* | Peonies occupy the spring breeze with their fragrance alone (Appendix D12). |
| 72 | 4 April 800 | Xi'an | Beginning date of fruit drop of *Salix babylonica* | The sycamore blooms after the willow catkins (Appendix D82). |
| 73 | 4 April 800 | Xi'an | First flowering date of *Firmiana platanifolia* | The sycamore blooms after the willow catkins (Appendix D82). |
| 74 | 4 April 800 | Xi'an | First flowering date of *Amygdalus davidiana* | Peach and plum flowers are fresh in every courtyard (Appendix D83). |
| 75 | 4 April 800 | Xi'an | First flowering date of *Prunus salicina* | Peach and plum flowers are fresh in every courtyard (Appendix D83). |
| 76 | 4 April 800 | Xi'an | First flowering date of *Paulowinia fortunei* | Paulownia blooms on Qingming Festival (Appendix D84). |
| 77 | 2 May 805 | Xi'an | End flowering date of *Paulowinia fortunei* | The purple paulownia flowers are falling and the birds are singing (Appendix D85). |
| 78 | 7 August 805 | Xi'an | First sing date of *Cryptotympana atrata* | A new cicada calls two or three times (Appendix D86). |
| 79 | 1 May 807 | Zhouzhi | End flowering date of *Paeonia suffruticosa* | When I come back, the peony flowers are all over (Appendix D87). |
| 80 | 10 June 807 | Zhouzhi | Beginning date of winter wheat harvest | People are busy in the fifth lunar month because the wheat is yellow |

| | | | | |
|---|---|---|---|---|
| | | | | in the field (Appendix D14). |
| 81 | 22 October 808 | Xi'an | First date of frost | Frost falls in the ninth lunar month and it turns cold early in autumn (Appendix D88). |
| 82 | 27 September 813 | Xi'an | Full-flowering date of *Osmanthus fragrans* | Osmanthus by the railing exudes fragrance (Appendix D89). |
| 83 | 13 May 815 | Xi'an | Beginning date of fruit drop of *Salix babylonica* | Willow catkins are flying all over the sky just like snowflakes (Appendix D90). |
| 84 | 3 April 820 | Xi'an | Full-flowering date of *Armeniaca vulgaris* | Although the apricot blossoms here are better than in other places, I still want to see the flowers in my hometown (Appendix D91). |
| 85 | 24 September 831 | Xi'an | Full-flowering date of *Osmanthus fragrans* | The cold dew wet the osmanthus quietly (Appendix D92). |
| 86 | 4 April 865 | Xi'an | End flowering date of *Armeniaca vulgaris* | Apricot flowers seem to be sad with me together (Appendix D93). |

**Appendix B: The modern data sources and reconstructing method in this study.**

Modern phenological observation data in Xi'an, which located in the center of Guanzhong Area, were derived from the China Phenological Observation Network (CPON). Xi'an has kept observations every year since 1963 except for the period of 1997–2002. The annual mean temperature data of 1951-2013 in Xi'an were obtained from the Chinese Meteorological Administration. Owing to a lack of
data, some modern phenophases were defined based on the meteorological data. For instance, the modern date of spring cultivation were defined as the first day when the daily mean temperature is consecutively higher than 5 ℃ for five days (Ge et al., 2010). The modern date of millet harvest in

autumn is defined as the first day when the daily mean temperature is continuously lower than 10 ℃ for

five days (Hao et al., 2009).

After changing the time series of temperature and phenophases to anomalies with respect to the

reference period (1961–1990 AD), the transfer functions between the phenological and temperature

anomalies were developed by linear regression, which can be expressed as:

$$y=ax_i+b \qquad\qquad\qquad (B1)$$

where $y$ is the annual temperature anomalies, and $x_i$ is the phenological anomalies for phenophase $i$. The

constants $a$ and $b$ are estimated using the least square method, which represents the regression slope and

intercept, respectively.

     Subsequently, the phenophase-specific transfer functions were applied to each historic phenological

anomaly to obtain the annual temperature anomalies. If there was more than one record in a single year,

temperature in that year was calculated as the arithmetic mean of all of the reconstructed temperatures

in that year.

**Appendix C: Transfer functions for the temperature reconstructions based on phenological records obtained from Liu et al (2016) and from poems in this study.**

| Phenophases | Transfer functions | Number of observations | Correlation coefficients | Standard error at 95% confidence level (℃) |
|---|---|---|---|---|
| First date of frost | $y=0.033x+0.423$ | 53 | 0.432[**] | 0.742 |
| Last date of frost | $y=-0.033x+0.386$ | 53 | -0.475[**] | 0.724 |
| First date of snow | $y=0.010x-0.023$ | 26 | 0.467[*] | 0.321 |
| Last date of snow | $y=-0.006x-0.019$ | 26 | -0.335 | 0.336 |
| First sing date of *Cryptotympana atrata* | $y=0.013x+0.012$ | 15 | 0.638 | 0.216 |
| Beginning date of spring cultivation | $y=-0.030x+0.232$ | 62 | -0.396[**] | 0.792 |

| | | | | |
|---|---|---|---|---|
| Beginning date of winter wheat harvest | $y=-0.084x+1.284$ | 22 | $-0.570^{**}$ | 0.584 |
| Beginning date of millet harvest | $y=0.024x+0.336$ | 61 | 0.231 | 0.806 |
| First flowering date of *Amygdalus davidiana* | $y=-0.075x+0.361$ | 38 | $-0.573^{**}$ | 0.667 |
| Full-flowering date of *Amygdalus davidiana* | $y=-0.086x+0.331$ | 38 | $-0.634^{**}$ | 0.630 |
| End flowering date of *Amygdalus davidiana* | $y=-0.069x+0.441$ | 37 | $-0.531^{**}$ | 0.691 |
| Fruit maturity date of *Amygdalus davidiana* | $y=0.022x+0.740$ | 13 | 0.495 | 0.505 |
| First flowering date of *Armeniaca mume* | $y=-0.044x+0.626$ | 14 | $-0.436$ | 0.785 |
| Full-flowering date of *Armeniaca mume* | $y=-0.055x+0.590$ | 14 | $-0.507$ | 0.752 |
| End flowering date of *Armeniaca mume* | $y=-0.061x+0.586$ | 14 | $-0.617^{*}$ | 0.717 |
| First flowering date of *Armeniaca vulgaris* | $y=-0.029x+0.119$ | 24 | $-0.320$ | 0.467 |
| Full-flowering date of *Armeniaca vulgaris* | $y=-0.045x+0.196$ | 20 | $-0.517^{*}$ | 0.402 |
| End flowering date of *Armeniaca vulgaris* | $y=-0.028x+0.135$ | 24 | $-0.331$ | 0.466 |
| First flowering date of *Chimonanthus praecox* | $y=-0.007x+0.669$ | 26 | 0.196 | 0.845 |
| Full-flowering date of *Chimonanthus praecox* | $y=-0.011x+0.770$ | 25 | $-0.218$ | 0.813 |
| First flowering date of | $y=-0.016x+0.135$ | 14 | $-0.217$ | 0.486 |

*Firmiana platanifolia*

| | | | | |
|---|---|---|---|---|
| First flowering date of *Hibiscus syriacus* | $y=-0.014x+0.060$ | 18 | -0.457 | 0.456 |
| Full-flowering date of *Juglans regia* | $y=-0.076x+0.441$ | 33 | -0.663* | 0.612 |
| Full-flowering date of *Osmanthus fragrans* | $y=-0.069x+0.306$ | 17 | -0.611** | 0.716 |
| End flowering date of *Osmanthus fragrans* | $y=0.044x+0.486$ | 22 | 0.497* | 0.728 |
| Full-flowering date of *Paeonia suffruticosa* | $y=-0.088x+0.307$ | 38 | -0.703** | 0.581 |
| End flowering date of *Paeonia suffruticosa* | $y=-0.065x+0.493$ | 36 | -0.446** | 0.731 |
| First flowering date of *Paulownia fortunei* | $y=-0.062x+0.688$ | 22 | -0.607* | 0.813 |
| End flowering date of *Paulownia fortunei* | $y=-0.055x+1.103$ | 18 | -0.382 | 0.901 |
| First flowering date of *Prunus salicina* | $y=-0.068x+0.585$ | 13 | -0.740** | 0.515 |
| Full-flowering date of *Prunus salicina* | $y=-0.068x+0.591$ | 13 | -0.779** | 0.480 |
| End flowering date of *Punica granatum* | $y=0.056x+0.257$ | 21 | -0.450 | 0.825 |
| Full-flowering date of *Pyrus betulaefolia* | $y=-0.076x+0.441$ | 27 | -0.698** | 0.608 |
| First leaf date of *Salix babylonica* | $y=-0.052x+0.745$ | 31 | -0.471** | 0.711 |
| Full leaf expansion date of *Salix babylonica* | $y=-0.042x+0.511$ | 37 | -0.384* | 0.753 |

| | | | | |
|---|---|---|---|---|
| Beginning date of fruit drop of *Salix babylonica* | $y=-0.091x+1.312$ | 17 | $-0.707^{**}$ | 0.602 |

*: $P<0.05$, **: $P<0.01$

**Appendix D: The original verses and sources of the poems in Chinese used in this paper.**

  1. "微月初三夜，新蝉第一声"（[唐]白居易《六月三日夜闻蝉》）；

2. "百泉冻皆咽，我吟寒更切"（[唐]刘驾《苦寒吟》）；

  3. "北风卷地白草折，胡天八月即飞雪"（[唐]岑参《白雪歌送武判官归京》）；

  4. "仍说秋寒早，年年八月霜"（[宋]司马光《晋阳三月未有春色》）；

  5. "乡村四月闲人少，才了蚕桑又插田"（[宋]翁卷《乡村四月》）；

  6. "梅信初传冬未深，高门熊梦庆相寻"（[宋]胡寅《吴守生朝》）；

7. "江涵秋影雁初飞，与客携壶上翠微"（[唐]杜牧《九日齐山登高》）；

  8."泛此黄金花，颓然清歌发"（[唐]李白《忆崔郎中宗之游南阳遗吾孔子琴抚之潸然感旧》）；

  9. "尊前柏叶休随酒，胜里金花巧耐寒"（[唐]杜甫《人日两首其二》）；

  10. "黄帝术，玄妙美金花"（[唐]吕岩《忆江南》其三）；

11. "赤霄玄圃须往来，翠尾金花不辞辱"（[唐]杜甫《赤霄行》）；

  12."澹荡韶光三月中，牡丹偏自占春风"（[唐]权德舆《和李中丞慈恩寺清上人院牡丹花歌》）；

  13."今年杜鹃花落子规啼，送春何处西江西"（[唐]白居易《送春归（元和十一年三月三十日作）》）；

14. "田家少闲月，五月人倍忙。夜来南风起，小麦覆陇黄"（[唐]白居易《观刈麦》）；

  15. "灞桥烟柳知何限，谁念行人寄一支"（[宋]陆游《秋夜怀吴中》）；

  16. "故园今日海棠开，梦入江西锦绣堆"（[宋]杨万里《春晴怀故园海棠二首》）；

  17. "碧鸡海棠天下绝，枝枝似染猩猩血"（[宋]陆游《海棠歌》）；

  18. "竹外桃花两三枝，春江水暖鸭先知"（[宋]苏轼《惠崇春江晚景》）；

19. "莱洲频度浅，桃实几成圆"（[唐]卢照邻《于时春也慨然有江湖之思寄赠柳九陇》）；

20. "人间四月芳菲尽，山寺桃花始盛开"（[唐]白居易《题大林寺》）；

21. "桃花欲落柳条长，沙头水上足风光。"（[唐]刘宪《上巳日祓褉渭滨应制》）；

22. "柳条弄色不忍见，梅花满枝空断肠"（[唐]高适《人日寄杜二拾遗》）；

23. "故人千万里，新蝉三两声"（[唐]白居易《立秋日曲江忆元九》）；

24. "要信今年春事早，社前十日燕新来"（[宋]陆游《新燕》）；

25. "学啭齐柳嫩，妍笑发春丛"（[唐]许敬宗《奉和登陕州城楼应制》）；

26. "接天莲叶无穷碧，映日荷花别样红"（[宋]杨万里《晓出净慈寺送林子方》）；

27. "向浦回舟萍已绿，分林蔽殿槿初红"（[唐]沈佺期《兴庆池侍宴应制》）；

28. "满街杨柳绿丝烟，画出清明二月天"（[唐]韦庄《鄜州寒食城外醉吟》）；

29. "曲池苔色冰前液，上苑梅香雪里娇"（[唐]崔日用《奉和人日重宴大明宫恩赐彩缕人胜应制》）；

30. "只为来时晚，花开不及春"（[唐]孔绍安《侍宴咏石榴》）；

31. "柳色迎三月，梅花隔二年"（[唐] 李百药《奉和初春出游应令》）；

32. "柳影冰无叶，梅心冻有花"（[唐] 李世民《冬日临昆明池》）；

33. "舒桃临远骑，垂柳映京营"（[唐]褚亮《奉和禁苑饯别应令》）；

34. "独有南山桂花发，飞来飞去袭人裾"（[唐]卢照邻《长安古意》）；

35. "月宫清晚桂，虹梁绚早梅"（[唐] 许敬宗《奉和过慈恩寺应制》）；

36. "上苑梅花早，御沟杨柳新"（[唐]骆宾王《西行别东台详正学士》）；

37. "柳色摇岁华，冰文荡春照"（[唐]卢照龄《七日登乐游故墓》）；

38. "梦梓光青陛，秾桃蔼紫宫"（[唐]刘祎之《奉和太子纳妃太平公主出降》）；

39. "映水轻苔犹隐绿，缘堤弱柳未舒黄"（[唐]马怀素《奉和立春游苑迎春应制》）；

40. "彩蝶黄莺未歌舞，梅香柳色已矜夸"（[唐]李显《立春日游苑迎春》）；

41. "梅香欲待歌前落，兰气先过酒上春"（[唐]卢藏用《奉和立春游苑迎春应制》）；

42. "林中觅草才生蕙，殿里争花并是梅"（[唐]沈佺期《奉和立春游苑迎春》）；

43. "剪绮裁红妙春色， 宫梅殿柳识天情"（[唐]崔日用《奉和立春游苑迎春应制》）；

44. "借问桃将李，相乱欲何如"（[唐]上官婉儿《奉和圣制立春日侍宴内殿出翦彩花应制》）；

45. "金阁妆新杏，琼筵弄绮梅"（[唐]宋之问《奉和立春日侍宴内出剪彩花应制》）；

46. "拂树添梅色，过楼助粉妍"（[唐]李峤《游禁苑陪幸临渭亭遇雪应制》）；

47. "今日回看上林树，梅花柳絮一时新"（[唐]赵彦昭《苑中人日遇雪应制》）；

48."山花缇绮绕，堤柳幔城开"（[唐]沈佺期《奉和晦日驾幸昆明池应制》）；

49."节晦莫全落，春迟柳暗催"（[唐]宋之问《奉和晦日幸昆明池应制》）；

50."野花飘御座，河柳拂天杯"（[唐]沈佺期《三日梨园侍宴》）；

51."泛桂迎尊满，吹花向酒浮"（[唐]李显《九月九日幸临渭亭登高得秋字》）；


52."寒尽梅犹白，风迟柳未黄"（[唐]宗楚客《正月晦日侍宴沪水应制赋得长字》）；

53."千行发御柳，一叶下仙筇"（[唐]张说《侍宴沪水赋得浓字》）；

54."绮萼成蹊遍簌芳，红英扑地满筵香"（[唐]李乂《侍宴桃花园咏桃花应制》）；

55."红萼竞燃春苑曙，荤茸新吐御筵开"（[唐]赵彦昭《侍宴桃花园咏桃花应制》）；

56."桃花灼灼有光辉，无数成蹊点更飞"（[唐]苏颋《侍宴桃花园咏桃花应制》）；


57."源水丛花无数开，丹跗红萼间青梅"（[唐]徐彦伯《侍宴桃花园》）；

58."林间艳色骄天马，苑里秾华伴丽人"（[唐]张说《桃花园马上应制》；

59."上阳柳色唤春归，临渭桃花拂水飞"（[唐]张说《奉和圣制初入秦川路寒食应制》）；

60."芳春桃李时，京都物华好"（[唐]崔湜《饯唐州高使君赴任》）；

61."香萼媚红滋，垂条萦绿丝"（[唐]徐彦伯《饯唐州高使君赴任》）；


62."宫梅间雪祥光遍，城柳含烟淑气浓"（[唐]阎朝隐《奉和圣制春日幸望春宫应制》）；

63."轻丝半拂朱门柳，细缬全披画阁梅"（[唐]李适《奉和春日幸望春宫应制》）；

64."光风摇动兰英紫，淑气依迟柳色青"（[唐]崔日用《奉和圣制春日幸望春宫应制》）；

65."晴风丽日满芳洲，柳色春筵祓锦流"（[唐]徐彦伯《上巳日祓禊渭滨应制》）；

66."宝马香车清渭滨，红桃碧柳禊堂春"（[唐]沈佺期《上巳日祓禊渭滨应制》）；


67."向浦回舟萍已绿，分林蔽殿槿初红"（[唐]沈佺期《兴庆池侍宴应制》）；

68."美人含遥霭，桃李芳自薰"（[唐]徐彦伯《题东山子李适碑阴二首》）；

69."独有归闲意，春庭伴落梅"（[唐]苏颋《和黄门舅十五夜作》）；

70."何当桂枝擢，还及柳条新"（[唐]张子容《长安早春》）；

71."暮春三月日重三，春水桃花满禊潭"（[唐]张说《三月三日定昆池奉和萧令得潭字韵》


）；

72."酒筵嫌落絮，舞袖怯春风"（[唐]王维《三月三日勤政楼侍宴应制》）；

73."杨花雪落覆白苹，青鸟飞去衔红巾"（[唐]杜甫《丽人行》）；

74."安得健步移远梅，乱插繁花向晴昊"（[唐]杜甫《苏端薛复筵简薛华醉歌》）；

75."千条嫩柳枝条垂拂青琐，百啭黄莺鸣叫声绕建章"（[唐]贾至《早朝大明宫呈两省僚友


》）；

76."五夜漏声催晓箭，九重春色醉仙桃"（[唐]杜甫《奉和贾至舍人早朝大明宫》）；

77."桃花细逐杨花落，黄鸟时兼白鸟飞"（[唐]杜甫《曲江对酒》）；

78."梨花度寒食，客子未春衣"（[唐]钱起《下第题长安客舍》）；

79."步屟随春风，村村自花柳"（[唐]杜甫《遭田父泥饮美严中丞》）；

80."春城无处不飞花，寒食东风御柳斜"（[唐]韩翃《寒食》）；

81."仲月风景暖，禁城花柳新"（[唐]李亨《中和节赐百官燕集因示所怀》）；

82."杨柳先飞絮，梧桐续放花"（[唐]元稹《咏廿四气诗 清明三月节》）；

83."深竹与清泉，家家桃李鲜"（[唐]权德舆《奉和崔阁老清明日候许阁老交直之际辱裴阁老书招云与考功苗曹长先城南游览独行口号因以简赠》）；

84."助君行春令，开花应清明"（[唐]白居易《答桐花》）；

85."怅望慈恩三月尽，紫桐花落鸟关关"（[唐]白居易《酬元员外三月三十日慈恩寺相忆见寄》）；

86."故人千万里，新蝉两三声"（[唐]白居易《立秋日曲江忆元九》）；

87."数日非关王事系，牡丹花尽始归来"（[唐]白居易《醉中归周至》）；

88."九月降霜秋早寒，禾穗未熟皆青乾"（[唐]白居易《杜陵叟》）；

89."画栏桂树悬秋香，三十六宫土花碧"（[唐]李贺《金铜仙人辞汉歌》）；

90."杨花榆荚无才思，惟解漫天作雪飞"（[唐]韩愈《晚春》）；

91."遮莫杏园胜别处，亦须归看傍村花"（[唐]王建《寒食忆归》）；

92."中庭地白树栖鸦，冷露无声湿桂花"（[唐]王建《十五夜望月》）；

93."鸥鸟似能齐物理，杏花疑欲伴人愁"（[唐]罗隐《清明日曲江怀友》）；