# Peer review of "How could phenological records from the Chinese poems of the Tang and Song Dynasties (618-1279 AD) be reliable evidence of past climate changes?"

_Climate of the Past, 2020_

## Referee Comment (RC1) · Anonymous Referee #1 · 9 Oct 2020

This paper examined the phenological records in Chinese poems during the 7-13th centuries, and tried to extract the climate information from these records. The scope falls within that of CP, and the topic is quite novel. It used scientific methods to interpret the phenological records in poems. The topic and methodology are new and open to discussions. It should be suitable for CP after major revision. My major concerns are, The dating of some poems are not very precise. Please try to find more poems that clearly recorded the dates. This could greatly improve the reliability of this research. The locations of some of the poems are not within the research area of this paper,

e.g., Appendix A: No 1,3,4,17,19. Maybe it is because the authors cannot find enough poems. However, if they can find more poems in the research area, the reliability of this research will also be improved significantly. As a whole, I would suggest the authors to try to find more records and expand the sample size. After supplement of relevant records, the reliability of the paper will be strengthened and merit publication in CP.

---

## Referee Comment (RC2) · Anonymous Referee #2 · 2 Nov 2020

As the author says, this paper provides a reference in both theory and method for the extraction and application of phenological records from poems in ancient China. This work would be valuable for the peers in the studies of past climate changes.

However,there are still some problems.(1)Chu (1973) laid the foundation for climate reconstructions based on documents. In his study, 17 pieces of evidence were from poems and 11 of them were phenological information of the Tang and Song Dynasties. In section 2, when the certainties and uncertainties of phenological information from poems are discussed, Chu's work would be a classic example. Specifically, which phenological information he extracted from poems was proved exact, and which was not, why? (2) In section 3.2, an important step should be added, which was the distinction between cultivated plant and wild plant. For example, some poems of late Tang dynasty referred that there were oranges planted in Xi'an, however, some researchers point out that these oranges were transplanted from southern China and couldn't overwinter normally in Guanzhong Plain.(3) Quan-Song-Shi (the Poetry of the Song Dynasty) is the main literature resource to reconstruct climate change during 960-1260 AD. However, in most period of 1127-1260 AD (Southern Song), North China war dominated by the Jin dynasty, so most of poems written by the poets living in North China during 1127-1260 AD are not contained in the Quan-Song-Shi. Are there some more literature sources?

---

## Author Comment (AC1) · 20 Nov 2020

Dear editors and reviewers,

Thank you very much for taking your time to review this manuscript. We really appreciate all your comments and suggestions which help us to improve our work. According to your comments, we make corresponding revisions to the original manuscript. Please find our point-to-point responses below.

Comment 1: The dating of some poems are not very precise. Please try to find more

poems that clearly recorded the dates. This could greatly improve the reliability of this research.

Response 1: Thank you for the suggestion. We have added an example (Line 181-185 of the revised manuscripïijǸsimilarly hereinafter), in which the temporal information of year, month, and date were detailed recorded in the poem. However, due to genre constraints, the lack of temporal information in ancient Chinese poetry is widespread. Some poems may have precise temporal information, while the writing time of most other poems was not consciously recorded. One of the highlights of our study is how to convert ambiguous temporal information recorded in the poems into precise phenological dates and make them participated in climate reconstructions. For example, the missing temporal information can be deduced according to the principle of conservative (Line 211-217) and can be deduced by consulting the background information (Line 310-319).

Comment 2: The locations of some of the poems are not within the research area of this paper, e.g., Appendix A: No 1,3,4,17,19.

Response 2: We are grateful for the comment. There are two kinds of phenological records from poems used in this study. One of them is used as examples to illustrate the characteristics of phenological records in poems and the handing methodology in the studies of past climate changes. The locations of this kind of poem (No.1-29 in Appendix D) are across China. The other kind of phenological records from poems are used as evidence to reconstruct the temperature anomalies in Guanzhong Area during 618-902 AD. The locations of the second kind of poems (No. 12, 14, 21, 29, 30-93 in Appendix D) are all in Guanzhong Basin. To eliminate potential misunderstandings, we have listed all 86 pieces of original records of the temperature reconstruction in Appendix A of the revised manuscript with Gregorian dates, sites, phenophases, and the translations of the original verses. Also, we have labeled the modern names of cities mentioned in the poems in Figure 2.

Comment 3: As a whole, I would suggest the authors to try to find more records and expand the sample size.

Response 3: Thanks for the comment and we would like to make some explanations. The primary goals of this paper are to prove that the phenological records from poems can be useful evidence of past climate changes and to provide a scientific processing methodology or the extraction and application of phenological records from poems. The temperature reconstruction of 618-902 AD in Guanzhong Area is a case study to demonstrate the validity of the processing methods mentioned in the paper. Our future work will focus on extracting more records from poems, and developing integration methods for different phenophases at different sites to explore the overall phenological change and climate change over a large region. In the future work we will try to provide a comparable study on how climate changed in the Tang and Song Dynasties.

---

## Author Comment (AC2) · 20 Nov 2020

Dear editors and reviewers, Thank you very much for taking your time to review this manuscript. We are grateful for the detailed suggestions, and we believe that these suggestions will considerably improve our paper. Please find our point-by-point responses below.

Comment 1: Chu (1973) laid the foundation for climate reconstructions based on documents. In his study, 17 pieces of evidence were from poems and 11 of them were

phenological information of the Tang and Song Dynasties. In section 2, when the certainties and uncertainties of phenological information from poems are discussed, Chu's work would be a classic example. Specifically, which phenological information he extracted from poems was proved exact, and which was not, why?

Response 1: Thank you for the suggestion. It's a great idea to introduce examples from world-renowned research to people who are not familiar with the field. We have added a few sentences (Line 92-103 of the revised manuscript, similarly hereinafter) to explain the certainties and uncertainties of phenological information from poems by introducing Chu's work in section 2.

Comment 2: In section 3.2, an important step should be added, which was the distinction between cultivated plant and wild plant. For example, some poems of late Tang dynasty referred that there were oranges planted in Xi'an, however, some researchers point out that these oranges were transplanted from southern China and couldn't overwinter normally in Guanzhong Plain.

Response 2: We are grateful for the suggestion. We have added a new subsection in 3.2.1 named "Filtering the records according to the human influence" (Line 268-274). And we have also redrawn Figure 1.

Comment 3: Quan-Song-Shi (the Poetry of the Song Dynasty) is the main literature resource to reconstruct climate change during 960-1260 AD. However, in most period of 1127-1260 AD (Southern Song), North China war dominated by the Jin dynasty, so most of poems written by the poets living in North China during 1127-1260 AD are not contained in the Quan-Song-Shi. Are there some more literature sources?

Response 3: Thanks for the comment. We agree with the point of view. Although some poets or scholars were active in the borderland China and ethnic minorities such as Yuan Haowen, most of the phenological records from poems of the Southern Song Dynasty we have at hand are located in southern China, especially around the city Hangzhou (the capital city of the Southern Song Dynasty). We will try to extract more

records from the Quan-Jin-shi (the poetry of the Jin Dynasty) to solve this problem in future work. We have added a few sentences to discuss this in Line 395-399.

---

## Referee Comment (RC3) · Anonymous Referee #3 · 28 Nov 2020

Review to the paper:

How could phenological records from the Chinese poems of the Tang and Song Dynasties (618-1260 AD) be reliable evidence of past climate changes?

Yachen Liu et al.

The paper raises the attention to an interesting and unique source of phenological information, early and high-medieval Chinese poems, and provides a preliminary analysis and temperature reconstruction referring to selected areas. It is well visible that

the authors invested much time and energy into this paper, the source is really unique and worth for further investigations. It is important to stress that I do appreciate many parts of the paper even if I do not mention them. However, in the review I rather try to point on the problems where in my opinion improvements are necessary. There are some basic methodological problems in the paper that has to be solved prior to publication: without a substantial improvement of the methodology, the paper is not suitable for publication in Climate of the Past. Therefore, I suggest major revision, and I would like to see and evaluate the next version of the manuscript.

Historical background and interpretation

The uncertainties are discussed in a rather detailed and informative way, and the authors also state that they only apply poems when the poets are contemporary – this is a very important and valuable information, what should be in my opinion emphasised also earlier (maybe already in the abstract?). Does this mean that the (contemporary) poets are known in all cases? And what cases are we talking about? Are these the poems where the 86 phenological data are coming from, or do the authors have a broader-scale overview, so that they are able to provide a general picture for a larger region than the study area (and if yes, based on how many poems/data)?

Just a side remark on the uncertainties subchapters: some of these uncertainties could be explained shorter and more accurately, once the authors involve a (Chinese) medieval social, economic or environmental historian as co-author of their study.

The authors discuss an over 600-year period covering the early and high-medieval period. Providing basic socio-economic background on how and why these poems were written (with reference), and the basic environmental characteristics (differences compared to recent times) of the environment the poets lived in and described should be an essential part of the presentation and analysis. As the topic is particularly sensitive on source dating, reliability and contemporary social/environmental background, the active participation of a trained (Chinese) medievalist, who can give a short concise

historical overview, would be in my opinion essential.

Geographical coverage

Even if it is clear that the authors would like to present the potentials of Chinese poems, and these potentials are valid for entire China, based on the information presented in line 305 and on their previous paper(s) in the subject, they have tested source potentials only in one area of one province. There is no problem with that but, please, do indicate this information at the beginning of the paper (i.e. you should have a "Study area" chapter at the beginning, which is a usual part of papers in CP), too. Because it is a rather important information that the authors do discuss this topic based on a database regarding entire China, but only one area within a province, and in fact you suggest that this might have relevance for the entire China. China is huge, and even in your study period there were long periods when China was not one empire, but an area divided to separate states. So, it would be also useful to discuss shortly why you think that in this rather eventful period of China's history this source was written in the same way and out of the same reasons when historical background (and also the level of literacy) in faraway regions could be rather different. Again, a (Chinese) medievalist would be able to answer this latter question easily and adequately.

Interpretation of past phenological information

The authors present both biological and physical phenological information. The biological information consists of plant and animal related phenological data. At first, I really needed to search a lot to figure out how many and what (wild) animal-related phenological data the authors actually used in the (case) study, and then I realised this was one bird type. It would be useful to state such information, because based on the main text (about source potentials of entire China and the entire study period) one expects several different types of animals. As for the plant-related phenological information, the authors mention different types: ornamental and cultivated plants.

What do you mean under "ornamental plant"? The only case where I saw any explanation was Table 1, where an example was added: "Plum blossoms begin to bloom in early winter". But plum is a fruit tree and as such, it is part of the cultivated vegetation, and fruit production is usually part of the agriculture. Why is it considered separately? Similarly, "ornamental animal" comes at one point in the picture, but it is not clear what it means and why it is mentioned.

I have some problems with the presentation of phenological information related to cultivated plants, as it seems the authors treat them as if they were similar modern cultivated plants. There is no any indication in the paper that early and high-medieval agriculture used rather different grain and other cultivated plant types/varieties (even plum or almond trees) than modern agriculture, not talking about the fact that medieval agriculture was on a totally different level than its modern equivalent. Although these differences usually have an effect on a temperature reconstruction, there is no any indication in the paper that the authors would have taken these differences into consideration. Again, the related knowledge of a Chinese historian expert would have basic importance. To some extent, the same is true for some of the physical indicators, particularly for the development of river ice (e.g. differences in streamflow due to river regulations, dams can strongly affect temperature-river ice relationship).

Moreover, it is not clear exactly what phenological phenomena the authors relate to what temperatures (i.e. what periods of the year), because the authors simply refer to Chinese Meteorological Administration, and do not give any further information. It would be useful to conclude shortly the information taken from these official records. I also have problem with using only 30 years (1961-1990) to identify the exact relationship between temperature (of what period?) and phenophase information. Phenology-based temperature reconstruction studies usually consider 50-60 years, at least, to identify this relationship. I understand that it is not possible to have longer overlap in some cases, but at least in those cases when it is possible to extend this control period, it would be useful to do it, and try out whether a longer control period gives the same relationship as 30 years.

Validation of results and Statistics

In the abstract, the authors refer to the abundance of the source (poems) and phenology information, but this abundance does not reflect on the applied database and the correlation statistics, where only 86 phenological data are available, covering only 38 years out of 300 years with any temperature-related information. Moreover, according to Appendix C, correlation statistics is based on a database where more than 2/3 of the phenological data types are calculated with the number of observations under 30, and 1/3 is under 20 – thus, in most cases the number of observations in fact does not reach the value to have any statistical significance. Moreover, sometimes even with the low observation number, correlations are rather low. In these cases, it would be useful to provide more information on why the authors think these data have further potentials. While in line 303 the authors suggest that they have selected 86 phenological records for validation, in line 382 the number of records is 85. So, is it 85 or 86? Either 85 or 86, this sounds like a rather low number for a reconstruction. Especially if we consider the fact that the authors used a number of different phenological data. I find the temperature reconstruction methodology a bit problematic. Based on Appendix B, in the reconstruction the authors applied the simple method of linear regression. However, in case of non-continuous datasets, as it is clearly the case with poem-based phenological information, the method of linear regression is not really a good method to apply. Could you explain why you think linear regression is the most suitable method to apply in this particular case? In fact (as I mentioned before), I also do not particularly like the fact that the authors treat this rather mixed set of early medieval phenological data automatically similar to those of the late 20th century.

I have read several times the validation subchapter and the related Appendix parts, but I still do not fully understand how the authors were able to reconstruct annual temperature anomalies. Do I understand well that – based on Fig. 3a, the Validation subchapter and the Appendices – the authors reconstructed annual temperature anomalies of over 300 years in a study area, based on 85 or 86 phenological data (if I understood well,

covering only 38 years)? How? This sounds far too little evidence for any temperature anomaly reconstruction. Such a temperature reconstruction would require that the database (near-)systematically cover the study period or at least a significantly higher number of observations. So, here a bit more explanation would be needed why the authors think 38 years of data can adequately describe the weather anomalies of 300 years.

In the Validation subchapter and in Fig. 3(b) the authors referred to another paper (Liu et al. 2016): this paper contains an annual temperature anomaly reconstruction for the period 600-902, in the Guanzhong Area – practically the same study area and period the current paper discusses. In Liu et al. 2016, the temperature reconstruction was based on 271 (phenological, weather and climate, and human response) data, from which 87 was phenological data. As we received little information on the exact 86 (or 85) phenological data the current study utilizes, the question arises whether or not there is an overlap of phenological data between the database of the current study and the phenology data part of the Liu et al. 2016 database. Especially, because the only phenological source quotation Liu et al. (2016) provides as an example is quoted from a poem. It is also not clear for me how and why this temperature reconstruction – or even the comparison with the Liu et al. 2016 paper – provides any validation for the utilisation potentials of poem-based phenological data. The authors used modern phenology-measured temperature relationship, applying it on early-medieval poem-based phenological data, to reconstruct early medieval annual temperature anomalies. As for the validation, as described above, it is not clear whether or not the Liu et al. (2016) reconstruction is independent from the current reconstruction. If not, the Liu et al. 2016 reconstruction should be applied with caution. Second: while comparing the two reconstructions in Fig. 3, the authors suggest that "There were approximately simultaneous temperature fluctuations between the two reconstructions,…" – well, looking at the Figure, this "simultaneous fluctuations" are not so easily and obviously recognisable. A statistically significant correlation would be a stronger proof for simultaneous fluctuation, but the authors do not provide any information on that. Dear

authors, please, give correlation data.

Accounting with so low data density and so many uncertainties, to me it seems somewhat surprising to state that annual temperatures were "0.43°C and 0.29°C higher during the study period (600-902 AD) than at present (1961-1990)." I doubt one can give such exact statements (without an estimation of uncertainties), when temperature related information is available only for 76 and 38 years out of 300 years. Based on these statements, I assume that the years for which information is not available were regarded as "average". However, if there is no poem referring to any phenophasis dates for 2-3 (or more) years in a row, this does not mean there could be no negative or positive temperature anomalies or even extremes in these years. It means only that no poem dealt with this question. In this respect, it would be useful to know how many different authors these 86 phenological data come from.

The authors do not compare their reconstruction to any other reconstructions from China. Is it because there are no other annually-resolved temperature reconstructions available in (Central-)China that cover the period 600-900? Because if there is at least one other, independent reconstruction (documentary based or natural scientific), then it would be useful to compare (and correlate) the current reconstruction results to that reconstruction (or reconstructions, if more than one exists).

And finally an addition: poems and songs are also applied in historical climatology in Europe, but it is not used independently for reconstruction, and poems very rarely contain phenological information (but it is not without an example).

---

## Author Comment (AC3) · 29 Dec 2020

Dear editors and reviewers,

Thank you very much for taking your time to review this manuscript. We are grateful for the detailed comments and suggestions, and we believe that these comments and suggestions will considerably improve our paper. Please find our point-by-point responses below.

Comment 1:

The uncertainties are discussed in a rather detailed and informative way, and the authors also state that they only apply poems when the poets are contemporary – this is a very important and valuable information, what should be in my opinion emphasized also earlier (maybe already in the abstract?). Does this mean that the (contemporary) poets are known in all cases? And what cases are we talking about? Are these the poems where the 86 phenological data are coming from, or do the authors have a broader-scale overview, so that they are able to provide a general picture for a larger region than the study area (and if yes, based on how many poems/data)?

Response 1:

Thank you for the comment. First, we would like to give brief introductions to the Quan-Tang-Shi (the poetry of the Tang Dynasty) and Quan-Song-Shi (the Poetry of the Song Dynasty), which are common sources for the poems of the Tang and Song Dynasties. Both Quan-Tang-Shi and Quan-Song-Shi are poetry collections of the two dynasties. The former was compiled in the Qing Dynasty (around 1705 AD), and the latter was compiled in modern times (after 1986). The titles, poets, and the verses were often recorded (in some cases, the titles and poets may also be unrecorded), but other information such as the writing time and places were not recorded in the Quan-Tang-Shi and Quan-Song-Shi. Just as we mentioned in Line 155-169 of the revised manuscript, the accessibility of phenological records of poems is relatively lower than that of other documents. For a specific poem, as we mentioned in Line 420-431, we cannot make sure whether it contains phenological information and is used in reconstructions before we read through the lines and related background information. Therefore, a standard procedure for extracting phenological records from poems is necessary to minimize the uncertainties of the records and filter out the useless records efficiently.

Our overall goals of this study are to demonstrate the validity and reliability of phenological records from poems as a proxy of past climate changes and to provide a reference in both theory and method for the extraction and application of phenological records from poems in China. Although we have only talked about the poems of the

Tang and Song dynasties and involved a case study of 86 phenological records for climate reconstruction, we believe that the methods of data extraction and processing are applicable for other areas and periods.

We have rewritten the abstract to make clear the objective of this study as well as explaining the source and function of the 86 phenological records. Please find the details in Line 24-33. To present a broader-scale overview, we have changed Section 2.2 into "The numbers, spatial distributions and accessibility of phenological records from poems". In this section, we have introduced the numbers of poems of the Tang and Song Dynasties (Line 144-147), the spatial distributions of phenological records of the Tang and Song Dynasties (Line 150-154) and the accessibility of phenological records from poems (Line 155-169).

Comment 2:

Just a side remark on the uncertainties subchapters: some of these uncertainties couldbe explained shorter and more accurately, once the authors involve a (Chinese) medieval social, economic or environmental historian as co-author of their study. The authors discuss an over 600-year period covering the early and high-medieval period. Providing basic socio-economic background on how and why these poems were written (with reference), and the basic environmental characteristics (differences compared to recent times) of the environment the poets lived in and described should be an essential part of the presentation and analysis. As the topic is particularly sensitive on source dating, reliability and contemporary social/environmental background, the active participation of a trained (Chinese) medievalist, who can give a short concise historical overview, would be in my opinion essential.

Response 2:

Thanks for the suggestion. Two authors of this manuscript, Xiuqi Fang and Junhu Dai, have expertise both in Chinese history and past environmental changes and are qualified to give a concise historical review. The Tang and Song Dynasties were two powerful and prosperous dynasties of imperial China. During these periods, society was relatively open, with Confucianism, Taoism and Buddhism coexisting. The developed economy made people more educated and could express their thoughts through literature. The status of literature, especially poetry, had also been elevated. The Imperial Examination System, a civil service examination system in imperial China for selecting candidates for the state bureaucracy, had gradually improved, and poetry was incorporated into the examination subjects during this period (Zhang 2015). People had the opportunities to gain attention and change their lives if they could write beautiful poems. In these contexts, as a literary genre, poetry reached its highest level during the Tang and Song Dynasties in ancient China. People in the Tang and Song Dynasties preferred to record their thoughts and daily lives in poems. The change in the Tang and Song Dynasties environment was controversial, spurring us to improve data resolution by extracting phenological records from poems for environmental reconstruction. We have also discussed this in Line 407-419.

We have added a paragraph to introduce the socio-economic background from the perspective of data sources. Please find the details in Line 71-79. We have also added another paragraph to introduce the environmental background from the perspective of climate reconstruction. Please find the details in Line 80-87.

Comment 3:

Even if it is clear that the authors would like to present the potentials of Chinese poems, and these potentials are valid for entire China, based on the information presented in line 305 and on their previous paper(s) in the subject, they have tested source potentials only in one area of one province. There is no problem with that but, please, do indicate this information at the beginning of the paper (i.e. you should have a "Study area" chapter at the beginning, which is a usual part of papers in CP), too. Because it is a rather important information that the authors do discuss this topic based on a database regarding entire China, but only one area within a province, and in fact you suggest that this might have relevance for the entire China. China is huge, and even

in your study period there were long periods when China was not one empire, but an area divided to separate states. So, it would be also useful to discuss shortly why you think that in this rather eventful period of China's history this source was written in the same way and out of the same reasons when historical background (and also the level of literacy) in faraway regions could be rather different. Again, a (Chinese) medievalist would be able to answer this latter question easily and adequately.

Response 3:

We are grateful for the suggestion and appreciate your profound understanding for China's history and related characteristics. As explained in Response 1, the Quan-Tang-Shi and Quan-Song-Shi were compiled according to the dynasties of the poets, and the spatial information was usually unknown in poems. In addition, the national border lines of the Tang and Song Dynasties had changed many times in history, but it did not affect the introduction of extracting and processing phenological records from poems. However, there were some features of the spatial distribution of phenological records from poems. The spatial distributions of phenological records were highly consistent with the ruling regions of the dynasties, and more developed areas had more records .

Furthermore, we do believe the methodologies put forward in this study are not only applicable to the Guanzhong Area, as you put forward kindly one area in one province, and the Tang Dynasty but also applicable to other periods and areas. The reasons are as follows. First, modern phenological studies have confirmed that the primary driving factor of phenophases in whole China is the climatic factors, especially the temperatures in most areas of temperate China (Chmielewski et al., 2001; Schwartz et al., 2006; Ge et al., 2015). Thus, the phenological records from poems can be used as indicators of climate changes for the places they obtained. Our work was to make the phenological records from poems meet the needs of quantitative reconstruction rather than change the expression of phenological phenomena. Second, historians all agreed that the feudal society in Chinese history had not fundamentally changed during different dynasties (Liu, 1981; Tian, 1982; Feng, 1994). The relatively stable feudal system is also the reason why the feudal society has lasted for more than 2000 years. Although historical China varied its border lines from dynasty to dynasty, its core social-economic closely aligned with the major agricultural area throughout history. This geographic and temporal overlap allows for continuous comparison across the Chinese core areas (Fang et al., 2019). Correspondingly, the essence of literature, especially poetry, has not changed, although different dynasties may have various characteristics of poetry such as the limitations on poetic forms, the number of words, the need for rhymes and sounds, etc.

We have made it clear in the Abstract (Line 29-33) that the methodologies proposed in this study were applicable for other periods and areas. The reasons were discussed in the Discussion section (Line 438-453). We have added the introduction of the spatial distributions of phenological records from poems in Section 2.2. Please find details in Line 150-154. In addition, in the case study, we have added a "Study area" Section to introduce the area of reconstruction. Please find details in Line 353-358.

Comment 4:

The authors present both biological and physical phenological information. The biological information consists of plant and animal related phenological data. At first, I really needed to search a lot to figure out how many and what (wild) animal-related phenological data the authors actually used in the (case) study, and then I realised this was one bird type. It would be useful to state such information, because based on the main text (about source potentials of entire China and the entire study period) one expects several different types of animals. As for the plant-related phenological information, the authors mention different types: ornamental and cultivated plants.

Response 4:

Thank you for the comment. There is abundant phenological evidence of different types of animals in the poems of the Tang and Song Dynasties. However, when it comes to

climate reconstruction, it is another story. On the one hand, as mentioned in Response 1, only when we read through all the poems of the Tang and Song Dynasties can we know how many types of animals there are. On the other hand, modern phenological observation in China focuses mainly on plants. Only a small number of early observational records refer to animal phenology. According to our historical and modern phenological data at hand, only the following animal phenology has the potential to participate in climate reconstruction: Cuculus micropterus (Indian Cuckoo), Oriolus chinensis (Black-naped oriole), Hirundo rustica (barn swallow) and Mogannia conica (a type of cicada).

There are two types of data in this study. One is used as examples to illustrate the characteristics of phenological records in poems and the handing methodology in the studies of past climate changes. The locations involved in this kind of poem (No.1-29 in Appendix D) are across China. Five of this kind of poem (No.1, 7, 13, 23, 24) described the phenology of 4 types of animals (cicada, geese, cuckoo and swallow). The other kind of phenological records are used as evidence to reconstruct the temperature anomalies in Guanzhong Area during 618-900 AD. The locations involved in this kind of poem (No. 12, 14, 21, 29, 30-93 in Appendix D) are distributed in the Guanzhong area. One of this kind of poem (No. 86) was related to the phenology of cicada. To eliminate potential misunderstandings, we have listed all 86 pieces of original records of the temperature reconstruction in Appendix A with Gregorian dates, sites, phenophases, and the translations of the original verses.

Comment 5:

What do you mean under "ornamental plant"? The only case where I saw any explanation was Table 1, where an example was added: "Plum blossoms begin to bloom in early winter". But plum is a fruit tree and as such, it is part of the cultivated vegetation, and fruit production is usually part of the agriculture. Why is it considered separately? Similarly, "ornamental animal" comes at one point in the picture, but it is not clear what it means and why it is mentioned.

Response 5:

Thanks for the comment. The phrase "ornamental plant" was used to express the concept as opposed to agricultural plants. We have changed the phrase to "natural plant" in the revised manuscript. The English word "plum" can refer to two different plants when translated into Chinese. One of them is "mei"(pronunciation in Chinese pinyin), which usually refers to Chimonanthus praecox or Armeniaca mume. They are the species we would like to express here. The other is "li", which usually refers to Prunus salicina, the species you understood here. To eliminate potential misunderstandings, we have changed the word to "ume" for the meaning of "mei". The word "plum" has remained for the translation of "li". Both "mei" and "li" in the Tang and Song Dynasties were natural plants because their phenophases were rarely affected by human activities at that time.

Comment 6:

I have some problems with the presentation of phenological information related to cultivated plants, as it seems the authors treat them as if they were similar modern cultivated plants. There is no any indication in the paper that early and high-medieval agriculture used rather different grain and other cultivated plant types/varieties (even plum or almond trees) than modern agriculture, not talking about the fact that medieval agriculture was on a totally different level than its modern equivalent. Although these differences usually have an effect on a temperature reconstruction, there is no any indication in the paper that the authors would have taken these differences into consideration. Again, the related knowledge of a Chinese historian expert would have basic importance. To some extent, the same is true for some of the physical indicators, particularly for the development of river ice (e.g. differences in streamflow due to river regulations, dams can strongly affect temperature-river ice relationship).

Response 6:

We are grateful for the comment and benefit by your expertise at this. We acknowledge the uncertainties caused by the difference in cultivated plant types and crop management. We would like to make some explanations . First, one of our data processing steps is "identifying the animals and plants to species level" (Section 3.2.2 in Line 297 to 315), which requires that the plants compared from modern observation and poems should be the same species. Second, modern phenological studies discuss a lot on the phylogenetic conservatism of phenology in response to climate changes, which has proved that phenological responses to climate changes are often shared among closely related species (Davies et al., 2013; Du et al., 2017; Davis et al., 2018). These studies indicate that even considering that evolution will lead to differences between historical and modern plants, the plants recorded in the poems and their corresponding modern observation plants still have similar responses to climate changes. Not to mention that a thousand years of time is too short for the evolutionary cycle of plants (Calderon, 1995; Donoghue, 2008). As for cultivated plants, many modern studies have proved that the phenophases of crops are mainly affected by climatic factors, especially temperature, compared with other factors such as agricultural management (Lobell et al., 2012; Tao et al., 2014; Liu et al., 2018). Third, though the traditional calibration procedure may make a contribution to this problem, it seems not suitable for our study. The common calibration procedure in climate reconstruction relies on the statistical calibration of climate proxy data against representative instrumental data based on data in a long period of overlap between the two datasets. However, the phenological data for producing phengological data series from poems are neither continuous nor from the same species. In addition, there is no overlapping period between phenological records from poems and observational data. For physical indicators, there is no record of ice phenology in our reconstruction of the Guanzhong Area (Appendix A).

We have added a short discussion about these uncertainties. Please find the details in Line 454-460.

Comment 7:

Moreover, it is not clear exactly what phenological phenomena the authors relate to what temperatures (i.e. what periods of the year), because the authors simply refer

to Chinese Meteorological Administration, and do not give any further information. It would be useful to conclude shortly the information taken from these official records. I also have problem with using only 30 years (1961-1990) to identify the exact relationship between temperature (of what period?) and phenophase information. Phenology based temperature reconstruction studies usually consider 50-60 years, at least, to identify this relationship. I understand that it is not possible to have longer overlap in some cases, but at least in those cases when it is possible to extend this control period, it would be useful to do it, and try out whether a longer control period gives the same relationship as 30 years.

Response 7:

Thank you for the comment. Many studies have concluded that the starting dates of the phenological phases are highly correlated with the temperature of the previous 2-3 months (Ahas et al., 2000; Piao et al., 2006; Dai et al., 2013). The 86 records in our study belonged to 34 different phenological phases corresponding to temperature in different periods of the year (Appendix A). However, in order to obtain a relatively uniform and comparable series of reconstructed temperatures, the mean annual temperature anomaly was selected as the reconstruction index. The correlation coefficients between the phenological phases and annual mean temperature were shown in Appendix C. We agree with you totally that the longer the time frame, the more accurate the relationship beteen phenology and a certain climatic variable will be. The period of 1961-1990 was selected as the reference period to calculate the mean annual temperature. By changing the time series of mean annual temperatures and phenological phases to anomalies with respect to 1961-1990, more data than 30 years were used to identify the exact relationship between anomalies of temperatures and phenophases. Taking the beginning date of spring cultivation as an example (Appendix C), as mentioned in Appendix B, the beginning date of spring cultivation was defined based on meteorological data, the data of 62 years (1951-2013, the data of 2006 was missing) were used to develop the transfer functions between anomalies of the beginning dates

of spring cultivation and mean annual temperatures. However, as we also mentioned in Appendix B (Line 771-772), the China Phenological Observation Network (CPON) began in 1963 and was stopped during 1997-2002. In addition, some phenophases may lack observations in specific years. Thus, some time series we used were less than 30 year but we have tried our best to use all available data.

We have rewritten the Data and Methods Section. The reason for the reconstructed period and the reference period we used were explained. Please find the details in Line 359-367.

Comment 8:

In the abstract, the authors refer to the abundance of the source (poems) and phenology information, but this abundance does not reflect on the applied database and the correlation statistics, where only 86 phenological data are available, covering only 38 years out of 300 years with any temperature-related information. Moreover, according to Appendix C, correlation statistics is based on a database where more than 2/3 of the phenological data types are calculated with the number of observations under 30, and 1/3 is under 20 – thus, in most cases the number of observations in fact does not reach the value to have any statistical significance. Moreover, sometimes even with the low observation number, correlations are rather low. In these cases, it would be useful to provide more information on why the authors think these data have further potentials. While in line 303 the authors suggest that they have selected 86 phenological records for validation, in line 382 the number of records is 85. So, is it 85 or 86? Either 85 or 86, this sounds like a rather low number for a reconstruction. Especially if we consider the fact that the authors used a number of different phenological data. I find the temperature reconstruction methodology a bit problematic. Based on Appendix B, in the reconstruction the authors applied the simple method of linear regression. However, in case of non-continuous datasets, as it is clearly the case with poem-based phenological information, the method of linear regression is not really a good method to apply. Could you explain why you think linear regression is the most suitable method to apply

in this particular case? In fact (as I mentioned before), I also do not particularly like the fact that the authors treat this rather mixed set of early medieval phenological data automatically similar to those of the late 20th century.

Response 8:

Thanks for your wise guidance at this point, challenging, but very constructive for the further modification of our manuscript. As mentioned in Section 2.2 "The numbers, spatial distributions and accessibility of phenological records from poems" (Line 141-169) and Response 1, the abundance of phenological evidence from poems is accompanied by relatively low data resolution for quantitative reconstruction. We have applied all modern and historical data at hand to the reconstruction. Besides, the reconstruction of the Guanzhong Area for the Tang Dynasty is just a case study to prove the validity of the quantitative reconstruction of past climate changes. The number of original records for quantitative reconstruction should be 86. Regarding the method of the linear regression, on the one hand, it is one of the common methods for reconstructions based on phenological data (Ge et al, 2003; Možná et al, 2012; Wetter et al., 2013). On the other hand, considering factors such as discontinuity of data, no overlapping period and limitation of modern data mentioned in Response 6 and 7, other methods such as process-based phenological model are either not applicable or would bring in more uncertainties. Although linear regression is not the most suitable reconstruction method but it is the only method we could use here. Our future work will focus on extracting more records from poems, and developing integration methods for different phenophases at different sites to explore the overall phenological change and climate change over larger regions. Regarding the comment about treating early medieval phenological data similar to those of the late 20th century, please find the explanations in Response 6.

Comment 9:

I have read several times the validation subchapter and the related Appendix parts, but

[Figure]

I still do not fully understand how the authors were able to reconstruct annual temperature anomalies. Do I understand well that – based on Fig. 3a, the Validation subchapter and the Appendices – the authors reconstructed annual temperature anomalies of over 300 years in a study area, based on 85 or 86 phenological data (if I understood well, covering only 38 years)? How? This sounds far too little evidence for any temperature anomaly reconstruction. Such a temperature reconstruction would require that the database (near-)systematically cover the study period or at least a significantly higher number of observations. So, here a bit more explanation would be needed why the authors think 38 years of data can adequately describe the weather anomalies of 300 years.

Response 9:

We are grateful for the comment pointed out doubts about the core outcome of our reconstruction. Per your guidance and suggestions, we have rewritten the part of the reconstruction and changed the title of chapter 4 into "Validation of the phenological records from poems for reconstructing the past climate changes: a case study of temperature reconstruction in the Guanzhong Area for specific years during 600-900 AD". Furthermore, the chapter has been further divided into three subchapters named 4.1 study area, 4.2 data and methods and 4.3 results and the comparisons with other reconstructions, in order to make the fact clearer. The reconstruction was introduced, analyzed and compared from the perspective of specific years instead of the whole period of 600-900 AD. Please find the details in Line 346-405.

Comment 10:

In the Validation subchapter and in Fig. 3(b) the authors referred to another paper (Liu et al. 2016): this paper contains an annual temperature anomaly reconstruction for the period 600-902, in the Guanzhong Area – practically the same study area and period the current paper discusses. In Liu et al. 2016, the temperature reconstruction was based on 271 (phenological, weather and climate, and human response) data, from

which 87 was phenological data. As we received little information on the exact 86 (or 85) phenological data the current study utilizes, the question arises whether or not there is an overlap of phenological data between the database of the current study and the phenology data part of the Liu et al. 2016 database. Especially, because the only phenological source quotation Liu et al. (2016) provides as an example is quoted from a poem. It is also not clear for me how and why this temperature reconstruction – or even the comparison with the Liu et al. 2016 paper – provides any validation for the utilisation potentials of poem-based phenological data. The authors used modern phenology-measured temperature relationship, applying it on early-medieval poembased phenological data, to reconstruct early medieval annual temperature anomalies. As for the validation, as described above, it is not clear whether or not the Liu et al. (2016) reconstruction is independent from the current reconstruction. If not, the Liu et al. 2016 reconstruction should be applied with caution. Second: while comparing the two reconstructions in Fig. 3, the authors suggest that "There were approximately simultaneous temperature fluctuations between the two reconstructions,. . ." –well, looking at the Figure, this "simultaneous fluctuations" are not so easily and obviously recognisable. A statistically significant correlation would be a stronger proof for simultaneous fluctuation, but the authors do not provide any information on that. Dear authors, please, give correlation data.

Response 10:

Thanks for the very kind and innovative suggestions. Although the study of Liu et al., 2016 was one of our previous works, it is independent of this study. In Liu et al., 2016, we obtained 87 phenological records (other records of weather, climate and human response were used to verify the results of temperature reconstruction) from diverse historical documents such as the Xin-Tang-Shu (New Book of Tang, the official history of the Tang Dynasty) to quantitatively reconstruct the winter half-year (from October to next April) temperatures in the Guanzhong Area from 600 to 902 AD. Except for one piece of data from a poem (No. 14 in Appendix A), there is no overlap between the

two databases. We believed that the reconstruction by Liu et al., 2016 is a good case for comparison with ours because of the same study area, the similar reconstruction period, the same data type from different sources, the similar data amounts (87 and 86), the same reconstruction index (We have obtained the original data from Liu et al., 2016 to reconstruct the mean annual temperature anomalies) and same transfer functions (For the same phenological evidence involved in both studies such as the first date of frost, they share the same transfer function). Thus, it proves the validation of phenological records from poems if the two studies have similar features in temperature variations. Meanwhile, the differences between the two reconstructions caused by the above factors could be eliminated. Regarding the modern phenology-measured temperature relationship on early-medieval poem-based phenological data, we have explained in Response 6. The statistically significant correlation between the two reconstructions was not applicable, because only a few reconstructed years of the two studies were overlapped.

We have reintroduced the reconstruction of Liu et al, 2016 and explained the reason why it was used as a comparison in Line 370-378. Moreover, we have also rewritten the comparison between our study and relevant reconstructions in Line 379-405.

Comment 11:

Accounting with so low data density and so many uncertainties, to me it seems somewhat surprising to state that annual temperatures were "0.43 and 0.29 higher during the study period (600-902 AD) than at present (1961-1990)." I doubt one can give such exact statements (without an estimation of uncertainties), when temperature related information is available only for 76 and 38 years out of 300 years. Based on these statements, I assume that the years for which information is not available were regarded as "average". However, if there is no poem referring to any phenophasis dates for 2-3 (or more) years in a row, this does not mean there could be no negative or positive temperature anomalies or even extremes in these years. It means only that no poem dealt with this question. In this respect, it would be useful to know how many

different authors these 86 phenological data come from.

Response 11:

Thank you for the comment and we are inspired by your questions. We have rewritten this part. The reconstructed temperature anomalies by phenological records from poems were treated as the temperature variations of specific years during the period of 600-900 AD. We also compared the occurrences of relatively cold and warm periods and the amplitudes of reconstructed temperatures with other relevant reconstructions. The uncertainties from transfer functions were shown in Figure 3 and Appendix C. Please find details in Line 391- 402. All the 86 pieces of original records of the temperature reconstruction have been listed in Appendix A and they belong to 69 poems from 39 poets.

Comment 12:

The authors do not compare their reconstruction to any other reconstructions from China. Is it because there are no other annually-resolved temperature reconstructions available in (Central-)China that cover the period 600-900? Because if there is at least one other, independent reconstruction (documentary based or natural scientific), then it would be useful to compare (and correlate) the current reconstruction results to that reconstruction (or reconstructions, if more than one exists).

Response 12:

Thanks for this very constructive suggestion and we made it clearer in this modification. As discussed in Line 407-419, the reconstructions based on natural evidence either cannot cover the whole period, or they have relatively low temporal resolutions. It is also the reason why we try to improve the spatiotemporal resolution of proxy by extracting phenological records from poems. We have added two relevant reconstructions for comparison. One of them was winter half-year temperature anomalies at a 30-year resolution reconstructed from documentary evidence in the middle and lower

reaches of the Yellow and Yangtze Rivers of China (Ge et al., 2003). The other was annual temperature anomalies reconstructed from tree rings in Asia (Ahmed et al., 2013). All the four reconstructions have been converted to temperature anomalies with respect to the mean climatology between 1961 and 1990 for comparison. Please find the details in Figure 3 and Line 393-405.

Comment 13:

And finally an addition: poems and songs are also applied in historical climatology in Europe, but it is not used independently for reconstruction, and poems very rarely contain phenological information (but it is not without an example).

Response 13:

Thank you for your confirmation of our main proxy data in this manuscript and the related comment. We really appreciate this. Although phenology and poems have been applied in historical climatology since Chu (1973), few studies have relied solely on phenological records or poetic content to reconstruct historical climate changes in China quantitatively. When we were finishing the work of Liu et al, 2016, we found that most of the phenological evidence in the traditional documents, such as the history books was non-organic. The idea of using poetry, which is the most popular literary form at that time, as a data source came to our minds. As mentioned in Line 120-140, poems in China contain abundant phenological evidence. However, as mentioned in Line 170-224 and Response 1, most of the essential information required for climate reconstruction such as the species, time and sites were hidden. That is also the reason why we try to provide a reference in both principle and methodology for the extraction and application of phenological records from poems.

References:

Ahas R, Jaagus J, Aasa A. The phenological calendar of Estonia and its correlation with mean air temperature. International Journal of Biometeorology, 2000, 44(4): 159-166.

Ahmed M, Anchukaitis K, Buckley B M et al. Continental-Scale Temperature Variability during the Past Two Millennia. Nature Geoscience, 2013, 6(5): 339-346.

Calderon W O . Phylogenetic Patterns among Tropical Flowering Phenologies. Journal of Ecology, 1995, 83(6):937-948.

Chmielewski F, Rotzer T. Response of tree phenology to climate change across Europe. Agricultural and Forest Meteorology, 2001, 108(2): 101-112.

Chu, K. A preliminary study on the climatic fluctuations during the last 5,000 years in China, Science China, 1973,16: 226-256.

Dai J H, Wang H J, Ge Q S. Multiple phenological responses to climate change among 42 plant species in Xi'an, China. International journal of biometeorology, 2013, 57(5): 749-758.

Davies T J, Wolkovich E M, Kraft N J B, et al. Phylogenetic conservatism in plant phenology. Journal of Ecology, 2013, 101(6): 1520-1530.

Davis C C , Willis C G , Primack R B, et al. The importance of phylogeny to the study of phenological response to global climate change. Philosophical Transactions of the Royal Society B: Biological Sciences, 2010, 365: 3201-3213.

Donoghue M J. A phylogenetic perspective on the distribution of plant diversity, Proceedings of the National Academy of Sciences, 2008, 105, 11549-11555.

Du Y, Chen J, Willis C G, et al. Phylogenetic conservatism and trait correlates of spring phenological responses to climate change in northeast China. Ecology  Evolution, 2017, 7: 6747–6757.

Fang X, Su Y, Wei Z et al. Social impacts of climate change in historical China, in: Socio-Environmental Dynamics along the Historical Silk Road, edited by: Yang L. E., Bork H.-R., Fang X., and Mischke S., Springer, Cham, Switzerland, 2019, 231-245.

Feng E. The evolution of Chinese social structure, Henan People's Publishing House,

Zhengzhou, China, 1994(in Chinese).

Ge Q S, Zheng J Y, Fang X Q et al. Winter half-year temperature reconstruction for the middle and lower reaches of the Yellow River and Yangtze River, China, during the past 2000 years. The Holocene, 2003, 13(6): 933-940.

Ge Q S, Wang H J, Dai J H. Phenological response to climate change in China: a meta-analysis. Global change biology, 2015, 21(1): 265-274.

Liu C: The reasons for the long-term continuation of Chinese feudal society, Historical Research, 1981, 2, 15-28 (in Chinese).

Liu Y, Chen Q, Ge Q et al. Spatiotemporal differentiation of changes in wheat phenology in China under climate change from 1981 to 2010. Science China Earth Sciences, 2018, 61: 1088–1097.

Lobell D B, Sibley A, Ivan Ortiz-Monasterio J. Extreme heat effects on wheat senescence in India. Nature Clim Change, 2012, 2: 186–189.

Možná M, Brázdil R, Dobrovolná P et al. Cereal harvest dates in the Czech Republic between 1501 and 2008 as a proxy for March–June temperature reconstruction. Climatic Change, 2012, 110(3-4):801-821.

Piao S L, Fang J Y, Zhou L M et al. Variations in satellite-derived phenology in China's temperate vegetation. Global Change Biology, 2006, 12(4): 672-685.

Schwartz M D, Ahas R, Aasa A. Onset of spring starting earlier across the Northern Hemisphere. Global Change Biology, 2006, 12(2): 343-351.

Tao F, Zhang Z, Xiao D et al. Responses of wheat growth and yield to climate change in different climate zones of China, 1981–2009. Agricultural and Forest Meteorology, 2014, 189: 91-104.

Tian J. A summary of the discussion on the reasons for the long-term continuation of Chinese feudal society, Historical Research, 1, 1982, 103-110 (in Chinese).

Wetter O, Pfister C. An underestimated record breaking event – why summer 1540 was likely warmer than 2003. Climate of the past, 2013, 9, 41-56.

Zhang, G. History of Chinese Poetry. Hebei Education Press, Shijiazhuang, China, 2015(in Chinese).

Please also note the supplement to this comment:
https://cp.copernicus.org/preprints/cp-2020-122/cp-2020-122-AC3-supplement.pdf

---

## Author Response (AR1)

**How could phenological records from the Chinese poems of the Tang and Song Dynasties (618-1260 AD) be reliable evidence of past climate changes?**

Yachen Liu1, Xiuqi Fang2, Junhu Dai3, Huanjiong Wang3, Zexing Tao3

- 1School of Biological and Environmental Engineering, Xi'an University, Xi'an, 710065, China
   2Faculty of Geographical Science, Key Laboratory of Environment Change and Natural Disaster MOE, Beijing Normal University, Beijing, 100875, China
   3Key Laboratory of Land Surface Pattern and Simulation, Institute of Geographic Sciences and Natural Resources Research, Chinese Academy of Science (CAS), Beijing, 100101, China
- 10 Correspondence to: Zexing Tao (taozx.12s@igsnrr.ac.cn)

**Abstract.** Phenological records in historical documents have been proved to be of unique value for reconstructing past climate changes. As a literary genre, poetry reached its peak period in the Tang and Song Dynasties (618-1260 AD) in China, which could provide abundant phenological records in this period when lacking phenological observations. However, the reliability of phenological records from

- 15 poems as well as their processing methods remains to be comprehensively summarized and discussed. In this paper, after introducing the certainties and uncertainties of phenological information in poems, the key processing steps and methods for deriving phenological records from poems and using them in past climate change studies were discussed: (1) two principles namely the principle of conservative and the principle of personal experience should be followed to reduce the uncertainties; (2) the phenological
- 20 records in poems need to be filtered according to the types of poems, the background information, the rhetorical devices and the spatial representations; (3) the animals and plants are identified to species level according to their modern distributions and the sequences of different phenophases; (4) the phenophases in poems are identified on the basis of modern observation criterion; (5) the dates and sites for the phenophases in poems are confirmed from background information and related studies. As a case study,
- 25 86 phenological records from poems of the Tang Dynasty in the Guanzhong Area of China were extracted to reconstruct the annual temperature anomalies in specific years of the period of 600-900 AD. Then the reconstruction from poems was compared with relevant reconstructions in published studies to demonstrate the validity and reliability of phenological records from poems in studies of past climate changes. This paper proved that the phenological records from poems could be useful evidence

30 of past climate changes after being scientifically processed and also provides a reference in both principle and methodology for the extraction and application of phenological records from poems not only for the study area and period in this study but also for larger areas and different periods in Chinese history.

**Keywords**. phenological records, poems, processing method, past climate changes, the Tang and Song Dynasties

**1** Introduction**

35

50

Phenology is the study of recurring biological life cycle stages and the seasonality of non-biological events triggered by environmental changes (Schwartz, 2003;Richardson et al., 2013). Phenological data derived from historical documents have been widely used as proxies to reflect past elimetic changes over the world, expectedly in Europe and Asia. The records of errors however dates

[revised manuscript text omitted]

---

## Author Response (AR2)

Dear editors and reviewers,

Thank you very much for taking your time to review this manuscript. We are grateful for the detailed suggestions, and we believe that these suggestions will make it clear for our paper. Please find our point-by-point responses below.

**1. Response to the anonymous reviewer 1**

Suggestion 1:

P4. Line 116.

Please modify "Tang Dynasty (618-902 AD)" to "Tang Dynasty (618-907 AD)". The Tang dynasty ended in 907 AD.

Response 1:

Thank you for the suggestion. We have changed the ending year to 907 AD of the Tang Dynasty in Line 116. And we have checked all the beginning and ending time of every dynasty mentioned in this paper.

Suggestion 2:

P2. Line 50

Please delete the second "the archives of"

Response 2:

We are grateful for the suggestion. Accordingly, we have deleted the second "the archives of" in Line 50.

Suggestion 3:

P23, Line 625

Mou C. Further research on the climatic fluctuations during the last 5000 years in China, China 625 Meteorological Press, Beijing, China, 1996(in Chinese).

Please double check the name of the author (牟重行) of this book. I checked the articles by him and found that, his name was spelled as Mu Zhongxing in some papers and Mou Chongxing in others. Please double check it.

Response 3:

Thanks for the suggestion and we would like to make some explanations. Actually, the three characters in this name are all polyphonetic characters in Chinese. The word "牟" has two pronunciations, "mou" and "mu" (pronunciations in Chinese pinyin). The two pronunciations of the word "重" are "chong" and "zhong". And the two pronunciations of the word "行" are "xing" and "hang". By confirming with the author, his name should be pronounced Mu Chongxing. We have revised the corresponding spelling in the text (Line 117) and references (Line 627).

Suggestion 4:

The authors mentioned Quan-Tang-Shi, Quan-Song-Shi, and Quan-Jin-Shi many times in the

paper, and these are actually also ancient Chinese books. Please add them in the references list and quote in the text.

Response 4:

We are grateful for the suggestion. Accordingly, we have quoted the Quan-Tang-Shi (Line 145), Quan-Song-Shi (Line 146), and Quan-Jin-Shi (Line 468-469) in the text and added them in the references (Line 647-648, Line 545-546 and Line 694).

**2. Response to the anonymous reviewer 2**

Suggestion 1:

Check the beginning and ending time of every dynasty in this paper. For example, in the title and abstract, the last year of Song dynasty is 1260AD, but in line 153, Song Dynasty lasted from 960 to 1279AD. "Tang Dynasty (618-902AD)" in line 116 is also incorrect. It is generally thought that the last year of Tang dynasty is 907AD.

Response 1:

Thank you for the suggestion. We have changed the ending year to 1279 AD of the Song Dynasty in the title (Line 2) and abstract (Line 13). And we have changed the ending year to 907 AD of the Tang Dynasty in Line 116. We also have checked all the beginning and ending time of every dynasty mentioned in this paper.

**3. Response to the editor**

Suggestion 1:

Thank you so much for submitting your revised version of manuscript. According to the comments, I see that there is quite a big improvement in your manuscript. Since the comments raised in this round are quite minor, I decide a minor revision for you. Please revise your manuscript accordingly.

Response 1:

We are really grateful to you and the reviewers for your quick processing and constructive suggestions. We have made the corresponding revisions according to the suggestions of the anonymous reviewers. We hope that these revisions in the manuscript and our related responses are sufficient to make our manuscript suitable for publication in Climate of the Past. We are looking forward to hearing from you at your earliest convenience.

---

## Author Response (AR3)

Dear editors,

Thank you very much for taking your time to review this manuscript. We are grateful for the latest suggestions. We have made the corresponding revisions. Please find our point-by-point responses below.

Suggestion 1:

Could you polish your language further?

Response 1:

Thank you for the suggestion, and we believe that this suggestion will improve the readability of the paper. We have invited a third-party company to polish the language in this paper. Please find details in the latest version of the manuscript.

Suggestion 2:

Could you check the format of your manuscript? Furthermore, could you use (Appendix A or B or C or D) when you refer to the content in Appendix? It will be much more clear.

Response 2:

We are grateful for the suggestion. Accordingly, we have changed numbers into Appendix D with numbers when we refer to the original lines of poems in the text. And we have checked other format issues of our paper according to the requirements of *Climate of the Past* and the latest articles published in the journal. Please find details in the latest version of the manuscript.